# AMPK-dependent and -independent coordination of mitochondrial function and muscle fiber type by FNIP1

**Liwei Xiao**[1☯], **Jing Liu**[1☯], **Zongchao Sun**[1☯], **Yujing Yin**[1], **Yan Mao**[1], **Dengqiu Xu**[1], **Lin Liu**[1], **Zhisheng Xu**[1], **Qiqi Guo**[1], **Chenyun Ding**[1], **Wanping Sun**[1], **Likun Yang**[1], **Zheng Zhou**[1], **Danxia Zhou**[1], **Tingting Fu**[1], **Wenjing Zhou**[2], **Yuangang Zhu**[2], **Xiao-Wei Chen**[2], **John Zhong Li**[3], **Shuai Chen**[4], **Xiaoduo Xie**[5], **Zhenji Gan**[1]\***

**1** MOE Key Laboratory of Model Animals for Disease Study, Department of Spine Surgery, Nanjing Drum Tower Hospital, The Affiliated Hospital of Nanjing University Medical School, Chemistry and Biomedicine Innovation Center (ChemBIC), Model Animal Research Center, Nanjing University Medical School, Nanjing University, Nanjing, China, **2** Institute of Molecular Medicine, Peking University, Beijing, China, **3** The Key Laboratory of Rare Metabolic Disease, Department of Biochemistry and Molecular Biology, The Key Laboratory of Human Functional Genomics of Jiangsu Province, Nanjing Medical University, Nanjing, China, **4** MOE Key Laboratory of Model Animals for Disease Study, Model Animal Research Center, Nanjing University Medical School, Nanjing University, Nanjing, China, **5** Department of Biochemistry, School of Medicine, Sun Yat-sen University, Shenzhen, China

☯ These authors contributed equally to this work.
\* ganzj@nju.edu.cn

**Data Availability Statement:** The RNA-seq data have been deposited in the NCBI Gene Expression Omnibus and are accessible through GEO Series accession number GSE150193.

## Abstract

Mitochondria are essential for maintaining skeletal muscle metabolic homeostasis during adaptive response to a myriad of physiologic or pathophysiological stresses. The mechanisms by which mitochondrial function and contractile fiber type are concordantly regulated to ensure muscle function remain poorly understood. Evidence is emerging that the Folliculin interacting protein 1 (*Fnip1*) is involved in skeletal muscle fiber type specification, function, and disease. In this study, *Fnip1* was specifically expressed in skeletal muscle in *Fnip1*-transgenic (*Fnip1*[Tg]) mice. *Fnip1*[Tg] mice were crossed with *Fnip1*-knockout (*Fnip1*[KO]) mice to generate *Fnip1*[TgKO] mice expressing *Fnip1* only in skeletal muscle but not in other tissues. Our results indicate that, in addition to the known role in type I fiber program, FNIP1 exerts control upon muscle mitochondrial oxidative program through AMPK signaling. Indeed, basal levels of FNIP1 are sufficient to inhibit AMPK but not mTORC1 activity in skeletal muscle cells. Gain-of-function and loss-of-function strategies in mice, together with assessment of primary muscle cells, demonstrated that skeletal muscle mitochondrial program is suppressed via the inhibitory actions of FNIP1 on AMPK. Surprisingly, the FNIP1 actions on type I fiber program is independent of AMPK and its downstream PGC-1α. These studies provide a vital framework for understanding the intrinsic role of FNIP1 as a crucial factor in the concerted regulation of mitochondrial function and muscle fiber type that determine muscle fitness.

**Funding:** This work was supported by grants from the National Natural Science Foundation of China (No. 91857105, 31771291 and 31922033 to Z.G. and 32071136 to T.F., www.nsfc.gov.cn), the Ministry of Science and Technology of China (National Key R&D Program of China 2018YFA0800700, www.most.gov.cn) and Natural Science Foundation of Jiangsu Province (BK20170014 and SWYY-002) to Z.G., State key Laboratory of Pharmaceutical Biotechnology, Nanjing University (KF-GN-202001) to X.X., Fundamental Research Funds for the Central Universities 090314380036 (to T.F.), 090314380031 and 090314380035 (to Z.G.). The funders had no role in study design, data collection and analysis, decision to publish, or preparation of the manuscript.

**Competing interests:** The authors have declared that no competing interests exist.

## Author summary

Mitochondria provide an essential source of energy to drive cellular processes and the function of mitochondria is particularly important in skeletal muscle, a metabolically demanding tissue that depends critically on mitochondria, accounting for ~40% of total body mass. In this study, we discovered an essential function of adaptor protein FNIP1 in the coordinated regulation of the mitochondrial and structural programs controlling muscle fitness. Using both gain-of-function and loss-of-function strategies in mice and muscle cells, we provide clear genetic data that demonstrate FNIP1-dependent signaling is crucial for muscle mitochondrial remodeling as well as type I muscle fiber specification. We also uncover that FNIP1 exerts control upon muscle mitochondrial program through AMPK but not mTORC1 signaling. Furthermore, we demonstrate that FNIP1 acts independently of PGC-1α to regulate fiber type specification. Hence, our study emphasizes FNIP1 as a dominant factor that coordinates mitochondrial and muscle fiber type programs that govern muscle fitness.

## Introduction

Mitochondria are essential organelles that serve diverse functions including bioenergetics, metabolism and signaling. Mitochondria are particularly important for maintaining metabolic homeostasis in skeletal muscle, the largest metabolic demanding tissue that undergoes extensive remodeling in response to myriad physiologic or pathophysiological stimuli. Significant evidence suggests that the function of mitochondria in skeletal muscle is compromised in the pathogenesis of many human diseases including obesity, type 2 diabetes and muscular dystrophy/atrophy [1–4]. Conversely, Exercise training is effective in counteracting the effects of many chronic diseases on mitochondrial function in skeletal muscle [4–8]. Thus, a better understanding of the molecular factors controlling mitochondrial function muscle fiber type programs could have implications for new therapeutic approaches for many human diseases associated with muscle mitochondrial defects.

Previous studies have demonstrated that the AMPK and PGC-1α are key transducers of exercise-mediated beneficial effects on muscle mitochondria. AMPK is activated by reduction of muscle ATP levels following exercise, providing a mechanism to integrate physiological exercise signals with the control of muscle mitochondrial system. AMPK was demonstrated to be essential for maintaining mitochondrial function in the skeletal muscle [9–11]. Indeed, AMPK activation in skeletal muscle is sufficient to drive a mitochondrial oxidative program in absence of any exercise [12,13]. AMPK-induced activation of muscle mitochondrial function is mediated, at least in part by PGC-1α signaling, which subsequently induces a broad array of genes involved in mitochondrial biogenesis and quality control through coactivating a number of nuclear receptors [14,15]. It is now clear that AMPK activates PGC-1α, likely through both direct phosphorylation of PGC-1α and promoting Sirt1-mediated PGC-1α activation [16,17].

Evidence is emerging that other factors may converge on AMPK and PGC-1α signaling, providing important layers of mitochondrial regulation in skeletal muscle. Folliculin interacting protein 1 (FNIP1) is an adaptor protein originally identified through its interaction with folliculin (FLCN) and AMPK [18]. Previous studies have implicated that FNIP1 influences cellular metabolic regulators such as AMPK and mTOR signaling [19–23]. However, the exact roles of FNIP1 in modulating AMPK or mTOR signaling remain unclear. Whole body deletion of *Fnip1* in mice has recently been shown to increase type I muscle fiber composition and activate AMPK signaling [20], whereas it has also been shown that mice with loss of FNIP1 in

muscle actually have reduced phosphorylation of the catalytic α subunit of AMPK (Thr172) [21]. Similarly, both suppression and activation of mTOR signaling by FNIP1 have also been reported [20,21,23,24]. Collectively, the results of previous studies suggest a role for FNIP1 in muscle fiber type specification and metabolism. However, whether and how FNIP1 orchestrate muscle mitochondrial and structural programs remain unclear.

We have recently found that MyomiR miR-499 regulates muscle mitochondrial respiration through inhibiting *Fnip1* [25]. Given the importance of AMPK-related signaling in mitochondrial biology and the potential role of FNIP1 in the metabolic basis of skeletal muscle function, we embarked on a study to define the precise control mechanism of FNIP1 signaling in skeletal muscle using both gain-of-function and loss-of-function strategies in mice. The markedly different muscle phenotypes of the FNIP1 deficiency mice and mice expressing FNIP1 only in skeletal muscle afforded us the unique opportunity to understand the intrinsic role of FNIP1 in the coordinated regulation of mitochondrial system and muscle fiber type that determine muscle function. Our results indicate that basal levels of FNIP1 are sufficient to simultaneously inhibit mitochondrial function and type I muscle fiber program in skeletal muscle. Using genetically engineered *Fnip1* and *Ampkα1/α2* triple knockout mouse models and primary skeletal myotubes in culture, we demonstrated that FNIP1 exerts control upon the skeletal muscle mitochondrial oxidative program through AMPK signaling. Surprisingly, the effects of FNIP1 on type I fiber program do not dependent on the AMPK/PGC-1α cascade. These findings provide a new mechanism by which mitochondrial function and muscle fiber type are concordantly regulated and have important implications for possible therapeutic manipulation of muscle mitochondrial function muscle fiber type for maintaining muscle fitness in a variety of chronic disease states.

## Results

### FNIP1-dependent regulation of mitochondrial remodeling in skeletal muscle

Muscle-specific *Fnip1* transgenic mice (*Fnip1*[Tg]) were established using the muscle creatine kinase promoter [26]. The Cas9 system targeting exon 6 of the *Fnip1* gene was also employed to generate *Fnip1* knockout mice (*Fnip1*[KO]). *Fnip1*[Tg] mice were crossed with *Fnip1*[KO] mice to generate *Fnip1*[TgKO] mice expressing *Fnip1* only in skeletal muscle but not in other tissues (Fig 1A). The mouse *Fnip1* transgene transcript was expressed in a skeletal muscle-specific manner, and we observed no overexpression of FNIP1 protein in the heart (S1A–S1C Fig). *Fnip1*[Tg] mice do not exhibit an overt muscle phenotype compared to nontransgenic (NTG) littermates (S1D Fig). In addition, *Fnip1* overexpression does not affect muscle fiber type specification (S1E and S1F Fig). However, loss of FNIP1 resulted in a pronounced change in the color of the skeletal muscle (S2 Fig), consistent with the previous observations [20]. Gene expression analyses confirmed the efficient knock out of *Fnip1* gene in *Fnip1*[KO] mice, while the mRNA expression of *Fnip2*, a close homolog of *Fnip1*, was increased in *Fnip1*[KO] muscle (S2C and S2D Fig). FLCN protein levels were also increased in *Fnip1*[KO] muscle (S2D Fig). As shown in Fig 1A, skeletal muscle specific restoration of *Fnip1* completely abolished the red coloration in *Fnip1*[KO] muscle (Fig 1A). Notably, *Fnip1*[KO] muscle weighted less than WT controls, and muscle weight normalized to body weight was also decreased in *Fnip1*[KO] mice compared to WT or *Fnip1*[TgKO] mice (S1 Table).

The muscle oxidative transformation is determined, largely in part, at the level of gene expression. To gain insight into the FNIP1-dependent regulation of skeletal muscle metabolism, transcriptome analysis was performed by whole-genome gene expression profiling experiments (GSE150193) in gastrocnemius (GC) muscle from both *Fnip1*[KO] and *Fnip1*[TgKO] mice.

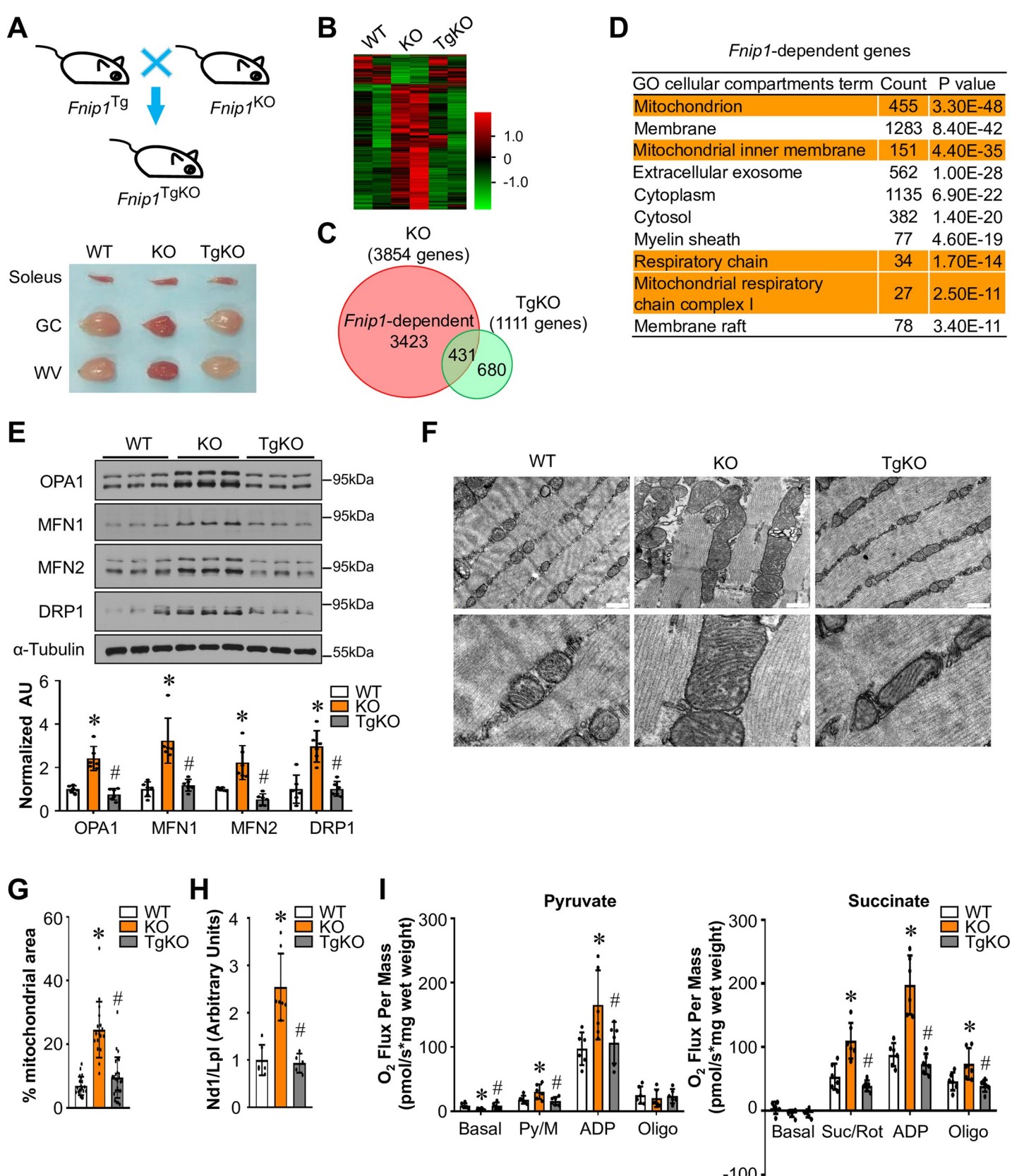

**Fig 1. FNIP1-dependent regulation of mitochondrial remodeling in skeletal muscle.** (**A**) (Top) Schematic showing the creation of *Fnip1*<sup>TgKO</sup> mice expressing *Fnip1* only in skeletal muscle but not in other tissues. (Bottom) Representative soleus, gastrocnemius (GC) and white vastus lateralis (WV) muscles from indicated mice at the age of 12 weeks. (**B**) Heat-map analysis of genes differentially regulated in *Fnip1*<sup>KO</sup> and *Fnip1*<sup>TgKO</sup> muscles. Each group is represented by RNA-Seq data from two independent samples generated from entire GC muscle of 8-week-old male mice, color scheme for fold change is provided. (**C**) Venn diagram comparing differential regulation of genes in *Fnip1*<sup>KO</sup> and *Fnip1*<sup>TgKO</sup> muscles. (**D**) Gene ontology (GO) enrichment analysis of gene transcripts (top 3000) regulated only in *Fnip1*<sup>KO</sup> but not in *Fnip1*<sup>TgKO</sup> muscles. (**E**) (Top) Representative immunoblot analysis of WV muscle lysates for mice of the indicated genotype using the indicated antibodies. (Bottom) Quantification of the OPA1/Tubulin, MFN1/Tubulin, MFN2/Tubulin and DRP1/Tubulin signal ratios were normalized (= 1.0) to the WT controls. n = 6 mice per group. (**F**) Representative electron micrographs of the tibialis anterior (TA) muscle showing intermyofibrillar (IMF) mitochondria in sections from wild-type (WT), *Fnip1*<sup>KO</sup> and *Fnip1*<sup>TgKO</sup> mice. The panels on the bottom show magnified areas of the panels on the top. The scale bar represents 500 nm. (**G**) Percentage of mitochondrial area per muscle fiber area was quantified from the indicated genotypes. 17–26 electron micrographs from each group were measured. (**H**) Results of quantitative PCR to determine mitochondrial DNA levels in TA muscle of the indicated mice using primers for NADH dehydrogenase (Nd1, mitochondria-encoded) and lipoprotein lipase (Lpl, nuclear-encoded). Nd1 levels were normalized to Lpl DNA content and expressed relative to WT (= 1.0) muscle. n = 5–7 mice per group. (**I**) Mitochondrial respiration rates were determined from the extensor digital longus (EDL) muscle of the indicated genotypes using pyruvate or succinate as substrates. Pyruvate/malate (Py/M) or succinate/rotenone (Suc/Rot)-stimulated, ADP-dependent respiration and oligomycin-induced (Oligo) are shown. n = 6 mice per group. Values represent mean ± SD, *$P < 0.05$ vs. WT controls, # $P < 0.05$ vs. *Fnip1*<sup>KO</sup>.

Knockout of *Fnip1* in skeletal muscle resulted in the up-regulation of 3115 genes and the down-regulation of 739 genes (S3 Fig), while genetic restoration of FNIP1 increased 846 genes and suppressed 265 genes (S4 Fig). Comparative analysis identified the induced expression of the major type I myosin *Myh7* and *Myh7b* genes paralleled the elevated expression of *miR-208b* and *miR-499* in *Fnip1*<sup>KO</sup> muscle (S3E and S3F Fig). Interestingly, the expression of the gene encoding *Esrrb* and *Esrrg*, known activators of *Myh7/miR-208b* and *Myh7b/miR-499* was also increased in *Fnip1*<sup>KO</sup> muscle compared with that in WT controls (S3E Fig). The up- or down-regulated genes are highlighted in red and green, respectively, in Fig 1B. We were particularly interested in the subset of genes that were regulated only in the *Fnip1*<sup>KO</sup> muscle but not in *Fnip1*<sup>TgKO</sup> muscle. The number of genes that were regulated uniquely or shared by the two genotypes is displayed in the Venn diagram (Fig 1C). Of the 3854 genes that were regulated in the *Fnip1*<sup>KO</sup> muscle, 3423 were uniquely regulated compared to the *Fnip1*<sup>TgKO</sup> muscle. In addition, 1111 genes were regulated in *Fnip1*<sup>TgKO</sup> muscle (Figs 1C and S4A). Gene ontology (GO) analysis revealed that the primary *Fnip1*-regulated genes were mitochondrial-related genes (Fig 1D). A broad array of genes involved in mitochondria oxidative program was regulated in *Fnip1*<sup>KO</sup> muscle in an *Fnip1*-dependent manner (Figs 1D and S3C and S2 Table). Together, the transcriptional profiling results indicate that a significant subset of *Fnip1* target genes in skeletal muscle, were those involved in mitochondrial program.

Validation studies confirmed that the expression of proteins involved in mitochondrial fusion and fission, including OPA1, MFN1, MFN2 and DRP1, was significantly up-regulated in the muscle of the *Fnip1*<sup>KO</sup> mice but reduced in *Fnip1*<sup>TgKO</sup> muscle (Figs 1E and S5A). These results suggested that FNIP1 are also critical for mitochondrial dynamics and quality control in skeletal muscle. We next analyzed mitochondrial morphology in wild-type (WT), *Fnip1*<sup>KO</sup> and *Fnip1*<sup>TgKO</sup> muscle. Mitochondrial volume density, size, and ultrastructure were assessed by electron microscopy (EM) in the intermyofibrillar (IMF) compartments of the tibialis anterior (TA) muscle fibers of all three genotypes (Fig 1F). EM analyses showed a striking increase of mitochondrial area per muscle fiber area in IMF regions in the *Fnip1*<sup>KO</sup> muscles compared to the other genotypes (Fig 1F and 1G). Notably, the WT and *Fnip1*<sup>TgKO</sup> mice have an ordered array of small IMF mitochondria along the Z-lines of the sarcomeres, whereas the *Fnip1*<sup>KO</sup> mice have more numerous and often significantly larger mitochondria, which displaced the myofibrils (Fig 1F). Consistent with the EM data, mitochondrial DNA levels were induced in parallel with the observed morphology changes among the genotypes (Fig 1H). Together, these results implicate FNIP1 in both muscle mitochondrial biogenesis and quality control programs.

Mitochondrial respiration rates were determined in muscle of all three genotypes using pyruvate or succinate as substrates (Fig 1I). Consistent with the morphology changes, both

pyruvate- and succinate-driven mitochondrial respiration rates were markedly induced in *Fnip1*<sup>KO</sup> muscles compared to the WT controls (Figs 1I and S5B). Mitochondrial respiration rates from *Fnip1*<sup>TgKO</sup> muscles were reduced to a level similar to WT controls (Fig 1I). Together, these data strongly demonstrate that FNIP1 is a negative regulator for skeletal muscle mitochondrial remodeling.

## FNIP1-dependent muscle oxidative fiber type transformation

The correction of muscle mitochondrial phenotypes in *Fnip1*<sup>TgKO</sup> mice afforded us the opportunity to use a comparative molecular profiling strategy to identify FNIP1-dependent regulatory pathway involved in the control of muscle metabolism. Consistent with the FNIP1-dependent regulation of mitochondrial biogenesis, immunoblotting analyses confirmed the increase of complex I (NDUFB8), complex III (UQCRC2), complex IV (COX4), and ATP synthase (ATP5A) proteins in *Fnip1*<sup>KO</sup> but not in *Fnip1*<sup>TgKO</sup> muscles (Figs 2A and S3C). The expression of myoglobin and cytochrome c, muscle oxidative metabolism biomarkers and known PGC-1α targets, was induced in *Fnip1*<sup>KO</sup> mice but completely reversed by *Fnip1* restoration (Figs 2B and S5C and S5D). The conversion between lactate and pyruvate in skeletal muscle is regulated by the LDHB/LDHA isoenzyme ratio, with LDHB favors the reaction that convert lactate to pyruvate and LDHA favors the reverse reaction to produce lactate [27,28]. The *Ldhb* gene expression was increased in *Fnip1*<sup>KO</sup> muscle but markedly reduced in *Fnip1*<sup>TgKO</sup> muscle (Fig 2C). The expected LDH isoenzyme activity shifts were confirmed by activity gel studies (Figs 2D and S5E). Moreover, muscle specific restoration of *Fnip1* reversed the reduction of blood lactate levels in *Fnip1*<sup>KO</sup> mice (Fig 2E).

Mitochondrial system is tightly coupled to muscle contractile fiber type. To further evaluate the effect of *Fnip1* on muscle oxidative transformation, we performed histochemical staining for succinate dehydrogenase (SDH) and myosin heavy chain (MHC) 1, which are hallmarks for oxidative metabolism, respectively [29]. As expected, the SDH enzymatic activities were higher in the GC muscle of *Fnip1*<sup>KO</sup> mice compared to WT controls or *Fnip1*<sup>TgKO</sup> mice (Fig 2F and 2G). MHC1 immunostaining demonstrated that *Fnip1* restoration reversed the induction of type I muscle fibers in both soleus and GC muscles of *Fnip1*<sup>KO</sup> mice (Figs 2F and 2G and S6A). These results further demonstrate a FNIP1-dependent muscle oxidative transformation.

## FNIP1 regulates AMPK but not mTORC1 in skeletal muscle cells

Recently, FNIP1 was shown to influence both AMPK and mTORC1 signaling in multiple cells [19–23], but the precise role of FNIP1 is uncertain. We thus examined the AMPK and mTOR signaling in the muscles of *Fnip1*<sup>KO</sup> and *Fnip1*<sup>TgKO</sup> mice. As shown in Fig 3A and 3B, Western blot confirmed that levels of phosphorylated AMPKα (Thr172) were higher in *Fnip1*<sup>KO</sup> but dramatically reduced in *Fnip1*<sup>TgKO</sup> muscle (Fig 3A and 3B). Whereas levels of phosphorylated mTOR as well as downstream products of mTORC1 activation, including phosphorylated ribosomal S6 protein kinase (p-S6K), ribosomal S6 protein (p-S6) and EIF4E-binding protein 1 (p-4EBP1), were not different in *Fnip1*<sup>KO</sup> muscles compared with *Fnip1*<sup>TgKO</sup> muscles (Figs 3A and 3B and S6B and S6C), suggesting that FNIP1 effect on mitochondrial oxidative program without changes in mTORC1 signaling in skeletal muscle. To further investigate the findings in a cell-autonomous setting, primary muscle myoblasts were isolated from GC muscles of *Fnip1*<sup>KO</sup> and WT mice, and then induced to differentiation for 3 days. As expected, *Fnip1*<sup>KO</sup> myotubes showed a significant increase in phosphorylation of AMPKα, and there are no changes in mTORC1 signaling in WT and *Fnip1*<sup>KO</sup> myotubes (Fig 3C and 3D). Subunits of

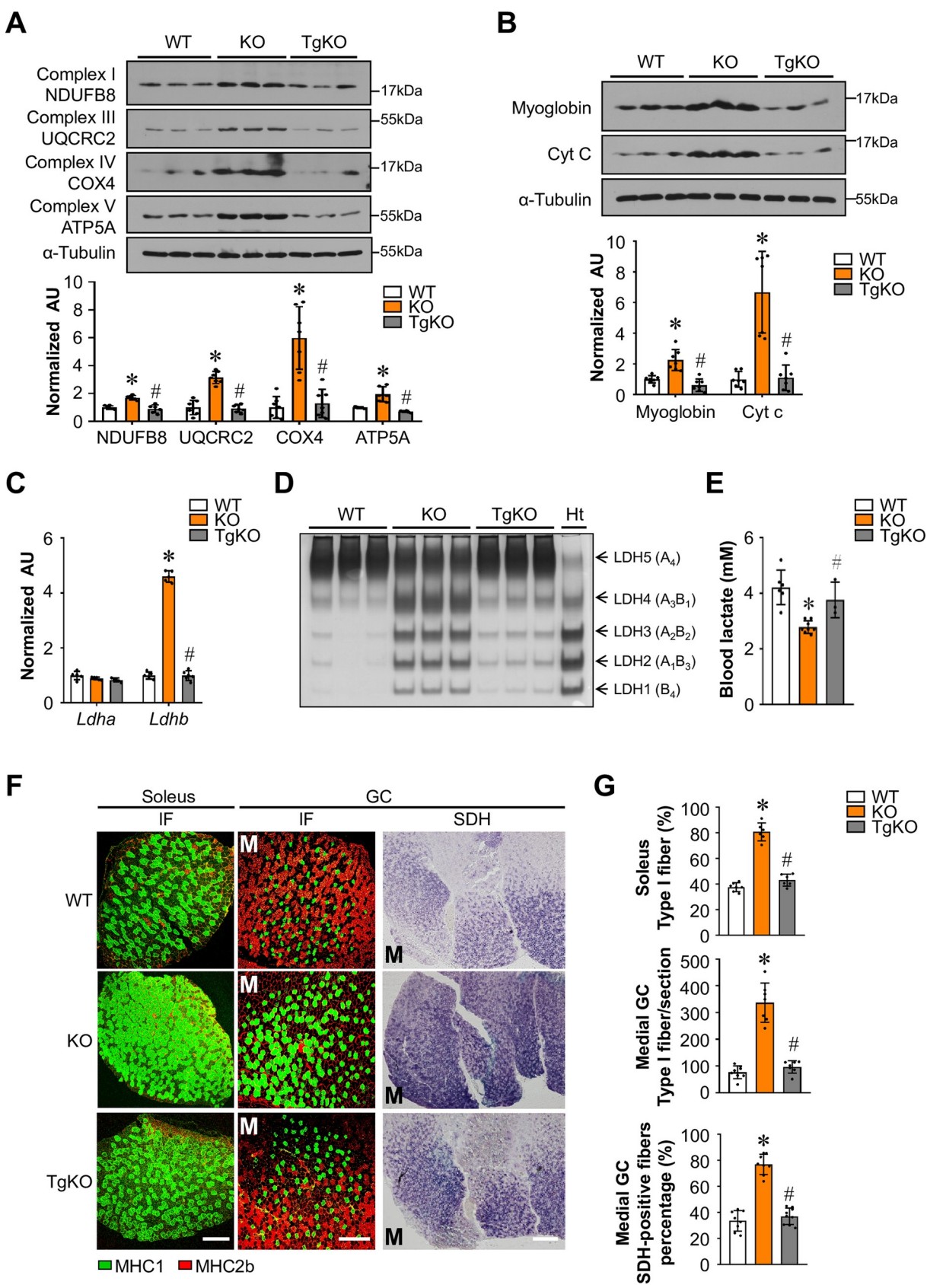

**Fig 2. FNIP1-dependent muscle oxidative fiber type transformation.** (**A**) (Top) Representative immunoblot analysis of entire GC muscle lysates for mice of the indicated genotype using the indicated antibodies. (Bottom) Quantification of the NDUFB8/Tubulin, UQCRC2/Tubulin, COX4/Tubulin and ATP5A/Tubulin signal ratios were normalized (= 1.0) to the WT controls. n = 5–7 mice per group. (**B**) (Top) Representative immunoblot analysis of WV muscle lysates for mice of the indicated genotype using the indicated antibodies. (Bottom) Quantification of the Myoglobin/Tubulin and Cytochrome c/Tubulin signal ratios were normalized (= 1.0) to the WT controls. n = 7 mice per group. (**C**) Results of RT-qPCR analysis of *Ldha* and *Ldhb* mRNA levels in entire GC muscles from indicated mice. n = 5–6 mice per group. (**D**) LDH isoenzymes were separated by polyacrylamide gel electrophoresis using whole cell extracts from heart (Ht, control) and WV muscle from the indicated mice. A representative gel showing 3 independent mice per group is shown. (**E**) Bars represent mean blood lactate levels from indicated male mice at the age of 8 weeks. n = 3–8 mice per group. (**F**) Cross-section of soleus and GC muscle from 8–9 weeks old male *Fnip1*^KO and *Fnip1*^TgKO mice stained for MHC immunofluorescence (IF) and SDH. Representative images were shown. M (medial head of GC). MHC1 (green), and MHC2b (red). Scale bar: 300 μm. (**G**) Quantification of IF and SDH data shown in (**F**). n = 6–9 mice per group. Values represent mean ± SD, \*$P < 0.05$ vs. WT controls, # $P < 0.05$ vs. *Fnip1*^KO.

the respiratory chain complexes and the ATP synthase were induced by *Fnip1*^KO in myotubes (Fig 3E).

To further determine the effects of FNIP1 on muscle cell mitochondrial function, oxygen consumption rates (OCR) were measured in primary myotubes. As shown in Fig 3F, knocking out of *Fnip1* significantly stimulated OCR in the presence of the uncoupler FCCP (Fig 3F). We also determined the extracellular acidification rate (ECAR) (a measure of glycolysis) along with OCR in these cells. *Fnip1*^KO significantly increased OCR/ECAR ratio in the presence of pyruvate, indicative a shift toward more oxidative phosphorylation for cellular energy production (Fig 3G). Taken together, these results demonstrate that *Fnip1* deficiency activates AMPK and mitochondrial oxidative program in skeletal muscle cells in a cell-autonomous manner.

## FNIP1 regulates muscle mitochondrial oxidative program through AMPK

We next addressed whether FNIP1 mediates its effect on skeletal muscle mitochondrial oxidative program through activation of AMPK. We bred *Fnip1*^KO mice with muscle-specific *Ampkα1/α2* double KO mice (*Ampkα1/α2*^f/f/Myf5-Cre) to generate *Fnip1*^KO, *Ampkα1/α2*^f/f/Myf5-Cre (TKO) mice, in which *Ampkα1/α2* genes are disrupted in muscle in *Fnip1*^KO background (Fig 4A). Muscle specific disruption of *Ampkα1/α2* resulted in marked reduction of pyruvate-driven mitochondrial respiration rates in the EDL muscle of *Fnip1*^KO mice (Fig 4B and 4C). Moreover, FNIP1 deficiency-mediated induction of mitochondrial dynamics proteins OPA1, MFN1 and MFN2, as well as oxidative biomarker cytochrome c was significantly reduced in the absence of *Ampkα1/α2* (Figs 4D and 4E and S7A). Furthermore, muscle specific disruption of *Ampkα1/α2* also resulted in decrease LDHB/LDHA ratio in *Fnip1*^KO muscle (Fig 4F). We also generated a second skeletal muscle-specific *Ampkα1/α2*-knockout mouse lines (*Ampkα1/α2*^f/f/MCK-Cre) using the muscle creatine kinase promoter-driven Cre [30]. As observed in TKO muscle, mitochondrial program was significantly decreased in *Fnip1*^KO, *Ampkα1/α2*^f/f/MCK-Cre mice, based on analysis of mitochondrial respiratory chain complexes and oxidative biomarkers in muscle (Fig 4G and 4H). Together with the results obtained from the TKO mice, these data strongly suggest that AMPK is required for the FNIP1 deficiency-mediated increase in mitochondrial oxidative program in skeletal muscle.

We also took advantage of primary muscle cell cultures derived from *Fnip1*^KO and TKO mice to further assess the AMPK-dependent regulation of mitochondrial program by FNIP1. Western blot confirmed that disruption of *Ampkα1/α2* completely abolish AMPK activation without affecting mTORC1 signaling in TKO myotubes (Fig 5A). We show that knocking out *Ampkα1/α2* blunted the effect of *Fnip1*^KO on the expression of the mitochondrial respiratory chain complexes and the ATP synthase, as well as the oxidative biomarkers myoglobin and cytochrome c (Figs 5B and S7B). Moreover, *Ampkα1/α2* ablation also resulted in marked reduction of mitochondrial dynamic gene expression in *Fnip1*^KO myotubes (Fig 5C).

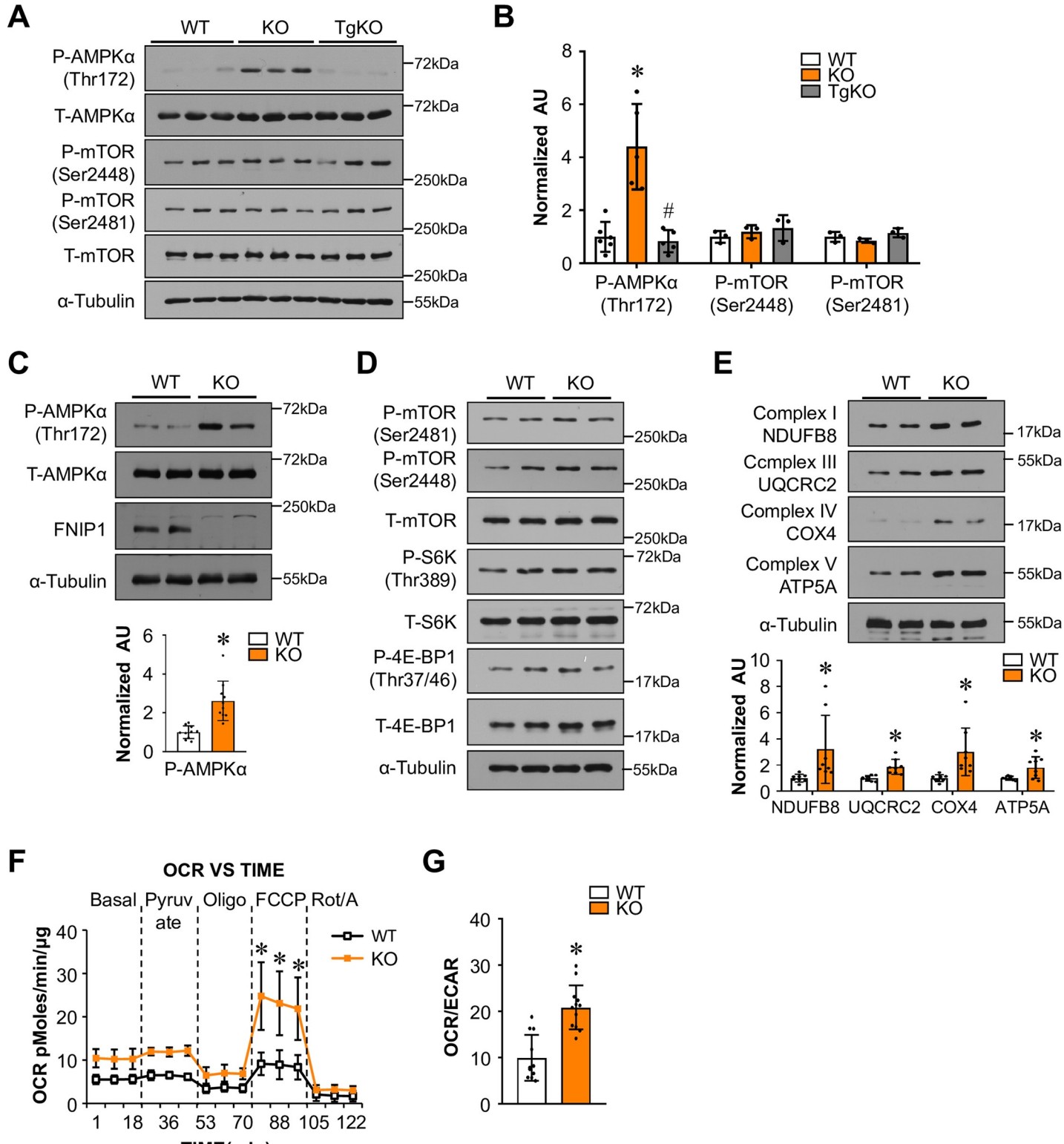

**Fig 3. FNIP1 inhibits muscle AMPK pathway and mitochondrial oxidative program in a cell-autonomous manner.** (**A**) Representative immunoblot analysis of WV muscle lysates for mice of the indicated genotype using the indicated antibodies. (**B**) Quantification of the p-AMPKα (Thr172)/AMPKα, p-mTOR (Ser2448)/mTOR, p-mTOR (Ser2481)/mTOR signal ratios were normalized (= 1.0) to the WT controls. n = 3–6 mice per group. (**C-E**) Primary skeletal muscle myoblasts were isolated from GC muscles of *Fnip1*^KO and WT mice, and then induced to differentiation for 3 days. (**C**) Immunoblot analysis of myotube extracts using the indicated antibodies.

Averaged p-AMPKα (Thr172)/AMPKα ratios were shown after normalization to the value of WT controls. n = 5 independent experiments. (**D**) Representative immunoblot analysis of myotube extracts using the indicated antibodies. n = 3 independent experiments. (**E**) Immunoblot analysis of myotube extracts using the indicated antibodies. Averaged NDUFB8/Tubulin, UQCRC2/Tubulin, COX4/Tubulin and ATP5A/Tubulin ratios were shown after normalization to the value of WT controls. n = 4 independent experiments. (**F**) Oxygen consumption rates (OCR) in primary myotubes harvested from indicated mice. Basal OCR was first measured, followed by administration of 10 mM sodium pyruvate, and 2 μM oligomycin (to inhibit ATP synthase), uncoupler FCCP (2 μM), or rotenone/antimycin (Rot/A; 1 μM) as indicted. n = 3 separate experiments done with 5–7 biological replicates. Data were analyzed by two-way ANOVA and Bonferroni post-hoc test. (**G**) OCR/ECAR ratio using pyruvate as substrate indicates a shift in cell energy production to oxidative phosphorylation. Values represent mean ± SD, $^{*}P < 0.05$ vs. WT controls, # $P < 0.05$ vs. $Fnip1^{KO}$.

Furthermore, mitotracker fluorescence analyses confirmed an increase in the mitochondrial mass in $Fnip1^{KO}$ myotubes compared to WT or TKO myotubes (S7C and S7D Fig). We also demonstrate that knocking out $Ampkα1/α2$ blunted the effect of $Fnip1^{KO}$ on mitochondrial OCR (Fig 5D and 5E). Taken together, these results demonstrate that the FNIP1/AMPK pathway regulates mitochondrial oxidative program in myotubes in a cell-autonomous manner.

## AMPK-independent regulation of muscle fiber type switching by FNIP1

We also investigate whether FNIP1 mediates its effect on muscle fiber type transformation through AMPK. Surprisingly, no significant changes in MHC1 immunofluorescence was observed in the soleus muscle of TKO mice compared to $Fnip1^{KO}$ mice ($Fnip1^{KO}$, 83.3 ± 6.2% vs. TKO, 82.2 ± 6.4%, n = 5–6 mice per group, p = 0.75) (Fig 6A and 6B). Similar observations were made in multiple muscle types of the TKO mice, in which the number of slow-twitch type I fibers was increased by $Fnip1^{KO}$ but not affected by disruption of $Ampkα1/α2$ in GC, and EDL muscle (Fig 6A and 6B). As observed in TKO muscle, no changes in type I fiber proportion was also observed in the GC muscle of $Fnip1^{KO}$ mice compared to $Fnip1^{KO}$, $Ampkα1/α2^{f/f/MCK-Cre}$ mice (Fig 6C). Consistent with the fiber typing results, gene expression studies demonstrate that the expression of the gene encoding the major slow-twitch type I myosin isoform MHC1 ($Myh7$ gene) and slow-twitch troponin genes was increased in both GC and soleus muscles of $Fnip1^{KO}$ mice but not affected by $Ampkα1/α2$ disruption (Figs 6D and S8). These data suggest that AMPK is indispensable for the FNIP1-mediated control of muscle mitochondrial system, but not fiber-type specification.

Previous studies have established that AMPK-induced activation of muscle mitochondrial function is mediated, at least in part by PGC-1α signaling [16,17]. To further confirm the role of AMPK signaling in FNIP1-mediated skeletal muscle mitochondrial reprogramming, we bred $Fnip1^{KO}$ mice with muscle-specific PGC-1α KO (PGC-1α mKO) mice to generate $Fnip1^{KO}$, PGC-1α$^{f/f/MCK-Cre}$ (DKO) mice, in which the PGC-1α gene is disrupted in muscle in $Fnip1^{KO}$ background (Fig 7A). Western blotting confirmed that the induced expression of PGC-1α protein was completely abolished in the DKO muscle (Fig 7B). As shown in Fig 7C, muscle specific disruption of PGC-1α resulted in marked reduction of pyruvate-driven mitochondrial respiration rates in the EDL muscle of $Fnip1^{KO}$ mice (Fig 7C). FNIP1 deficiency-mediated induction of mitochondrial genes (NDUFB8, ATP5A, and cytochrome c) was significantly reduced in the absence of PGC-1α (Figs 7D and S9). Moreover, PGC-1α ablation also resulted in marked reduction of mitochondrial dynamic gene expression in $Fnip1^{KO}$ muscle (Fig 7E). Furthermore, muscle specific disruption of PGC-1α also reversed LDHB/LDHA isoenzyme ratio in $Fnip1^{KO}$ muscle (Fig 7F). Together, these results further establish the importance of AMPK/PGC-1α cascade in the FNIP1 deficiency-mediated activation of mitochondrial oxidative program in skeletal muscle.

SDH histochemical staining reveled that muscle specific disruption of PGC-1α resulted in marked reduction of mitochondrial oxidative activity in the GC muscle of $Fnip1^{KO}$ mice (Fig 7G). Interestingly, however, no significant changes in MHC1 immunofluorescence was

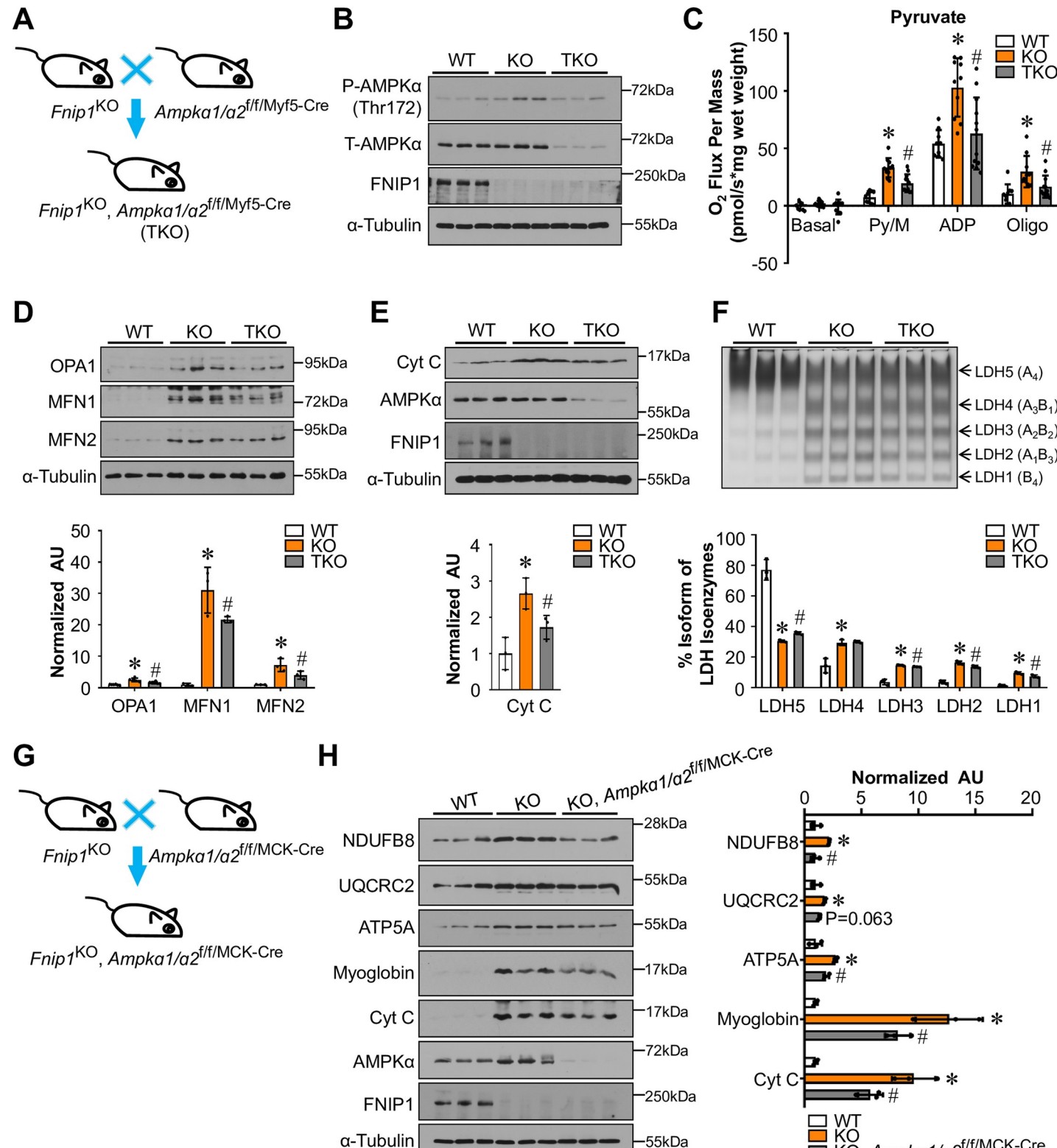

**Fig 4. FNIP1 regulates muscle mitochondrial oxidative programs through AMPK. (A)** Schematic showing the creation of *Fnip1*^KO, *AMPKa1/a2*^f/f/Myf5-Cre (TKO) mice. (**B**) Representative immunoblot analysis performed with EDL muscle total protein extracts prepared from the indicated mice using p-AMPKα (Thr172), AMPKα, FNIP1 and α-Tubulin (control) antibodies. n = 6 mice per group. (**C**) Mitochondrial respiration rates were determined from EDL muscle of the indicated genotypes using pyruvate as a substrate. Pyruvate/malate (Py/M)-stimulated, ADP-dependent respiration and oligomycin-induced (Oligo) are shown. n = 9–12 mice per group.

(**D**, **E**) (Top) Representative immunoblot analysis of TA muscle lysates for mice of the indicated genotype using the indicated antibodies. (Bottom) Quantification of the OPA1/Tubulin, MFN1/Tubulin, MFN2/Tubulin and Cytochrome c/Tubulin signal ratios were normalized (= 1.0) to the WT controls. n = 3–6 mice per group. (**F**) (Top) LDH isoenzymes were separated by polyacrylamide gel electrophoresis using whole cell extracts from EDL muscle from the indicated mice. A representative gel showing 3 independent mice per group is shown. (Bottom) Quantification of LDH isoenzyme activity gel electrophoresis was shown. (**G**) Schematic showing the creation of *Fnip1*$^{KO}$, *AMPKa1/a2*$^{f/f/MCK-Cre}$ mice. (**H**) (Left) Immunoblot analysis of WV muscle lysates for mice of the indicated genotype using the indicated antibodies. (Right) Quantification of the NDUFB8/Tubulin, UQCRC2/Tubulin, ATP5A/Tubulin, Myoglobin/Tubulin and Cytochrome c/Tubulin signal ratios were normalized (= 1.0) to the WT controls. n = 3 mice per group. Values represent mean ± SD, *$P$ < 0.05 vs. WT controls, # $P$ < 0.05 vs. *Fnip1*$^{KO}$.

observed in both soleus and GC muscle of DKO mice compared to *Fnip1*$^{KO}$ mice (Fig 7H and 7I). These results demonstrate that PGC-1α is dispensable for the FNIP1 deficiency-mediated control of type I muscle fiber formation, which are consistent with AMPK-independent regulation of muscle fiber type specification by FNIP1.

## Discussion

Skeletal muscle comprises ~40% of human body mass and is a key determinant of whole body energy metabolism. Mitochondria are undoubtedly critical for muscle functions. The molecular mechanisms involved in the coordinated regulation of the mitochondrial and structural determinants of muscle function are of particular interest because of their relevance to many human disorders, including obesity, type 2 diabetes and muscular diseases [1–4]. In this study, we identify FNIP1 as a crucial factor that dually regulates skeletal muscle mitochondrial and structural programs through both AMPK-dependent and -independent mechanisms. Our findings implicate the FNIP1 signaling serves as a dominant pathway that coordinates mitochondrial system and skeletal muscle fiber type that determine muscle fitness.

Our findings that muscle FNIP1 orchestrates mitochondrial function and type I muscle fiber programs through both AMPK-dependent and -independent mechanisms revealed an important intrinsic role of FNIP1 in skeletal muscle metabolic control, and elucidated a mechanism by which mitochondrial function and muscle fiber type are concordantly regulated. Previous study using whole body knockout mice has provided a clue that FNIP1 is involved in skeletal muscle fiber type specification, function, and disease [20]. However, whole body deletion of *Fnip1* also leads to multiple nonmuscle system derangements including immune cell development defect, cardiomyopathy, and renal cyst formation [19,20,22,31,32]. It is unclear whether the muscle changes are autonomous to muscle cells or secondary effects driven by deletion of *Fnip1* in nonmuscle tissues. Moreover, although previous studies have implicated that FNIP1 could influence both AMPK and mTORC1 signaling in multiple cell types [19–23], reports on the effects of FNIP1 on AMPK and mTORC1 signaling have been inconsistent and, thus, less conclusive [19–24]. Therefore, the intrinsic physiological role of FNIP1 in skeletal muscle remains unclear. In this study, we took advantage of the *Fnip1*$^{KO}$ and *Fnip1*$^{TgKO}$ mice expressing *Fnip1* only in skeletal muscle but not in other tissues to explore the detailed insights of FNIP1 actions in skeletal muscle. Gain-of-function and loss-of-function studies in mice, together with assessment of primary skeletal myotubes in culture, afforded us the unique opportunity to define the intrinsic role of FNIP1 in skeletal muscle. Our strong genetic data led to the following surprising conclusions: 1) basal levels of FNIP1 are sufficient to function as a "brake" to simultaneously inhibit mitochondrial and type I fiber programs in skeletal muscle, while muscle-specific *Fnip1* transgenic mice (*Fnip1*$^{Tg}$) appeared normal on inspection, and did not exhibit a muscle phenotype compared to NTG littermates; 2) FNIP1 in fact cell-autonomously regulates AMPK but not mTORC1 signaling in skeletal muscle cells, thus shed light on the inconsistencies in previous data; 3) FNIP1 regulates muscle mitochondrial oxidative program through AMPK, these findings are recapitulated in skeletal muscle cell culture and, thus, are cell autonomous; 4) FNIP1 regulates type I fiber program independent of

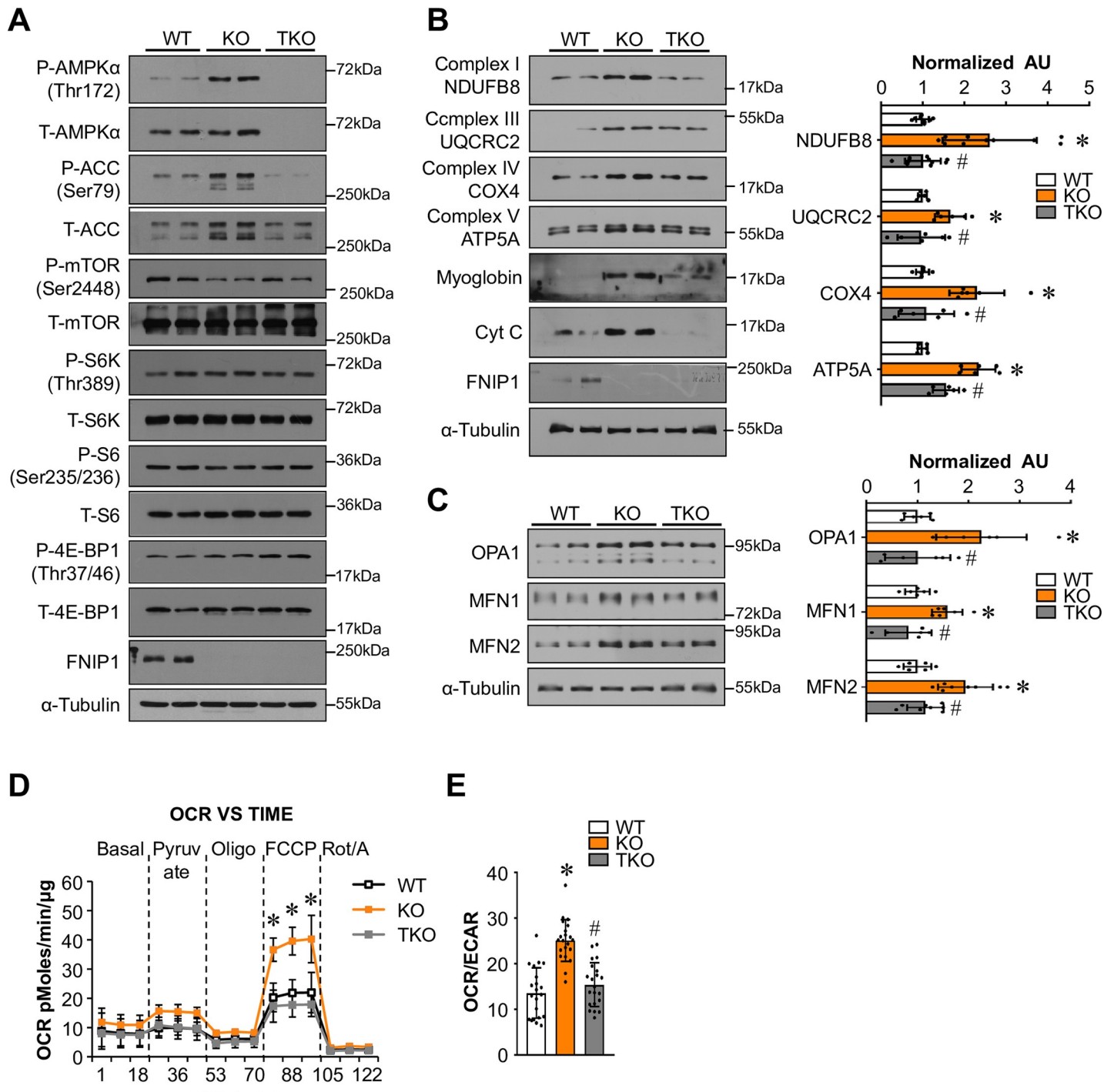

**Fig 5. FNIP1 regulates muscle mitochondrial oxidative programs through AMPK in a cell-autonomous manner.** (**A-C**) Primary myoblasts were isolated from GC muscles of indicated mice, and then induced to differentiation for 3 days. (**A**) Representative immunoblot analysis of myotube extracts using the indicated antibodies. n = 3–4 independent experiments. (**B**, **C**) Immunoblot analysis of myotube extracts using the indicated antibodies. Averaged NDUFB8/Tubulin, UQCRC2/Tubulin, COX4/Tubulin, ATP5A/Tubulin, OPA1/Tubulin, MFN1/Tubulin and MFN2/Tubulin ratios were shown after normalization to the value of WT controls. n = 3–5 independent experiments. (**D**) Oxygen consumption rates (OCR) in primary myotubes harvested from indicated mice. Basal OCR was first measured, followed by administration of 10 mM sodium pyruvate, and 2 μM oligomycin (to inhibit ATP synthase), uncoupler FCCP (2 μM), or rotenone/antimycin (Rot/A; 1 μM) as indicted. n = 3 separate experiments done with 5–7 biological replicates. Data were analyzed by two-way ANOVA and Bonferroni post-hoc test. (**E**) OCR/ECAR ratio using pyruvate as substrate indicates a shift in cell energy production to oxidative phosphorylation. n = 3 separate experiments done with 5–7 biological replicates. Values represent mean ± SD, $^{*}P < 0.05$ vs. WT controls, # $P < 0.05$ vs. *Fnip1*[KO].

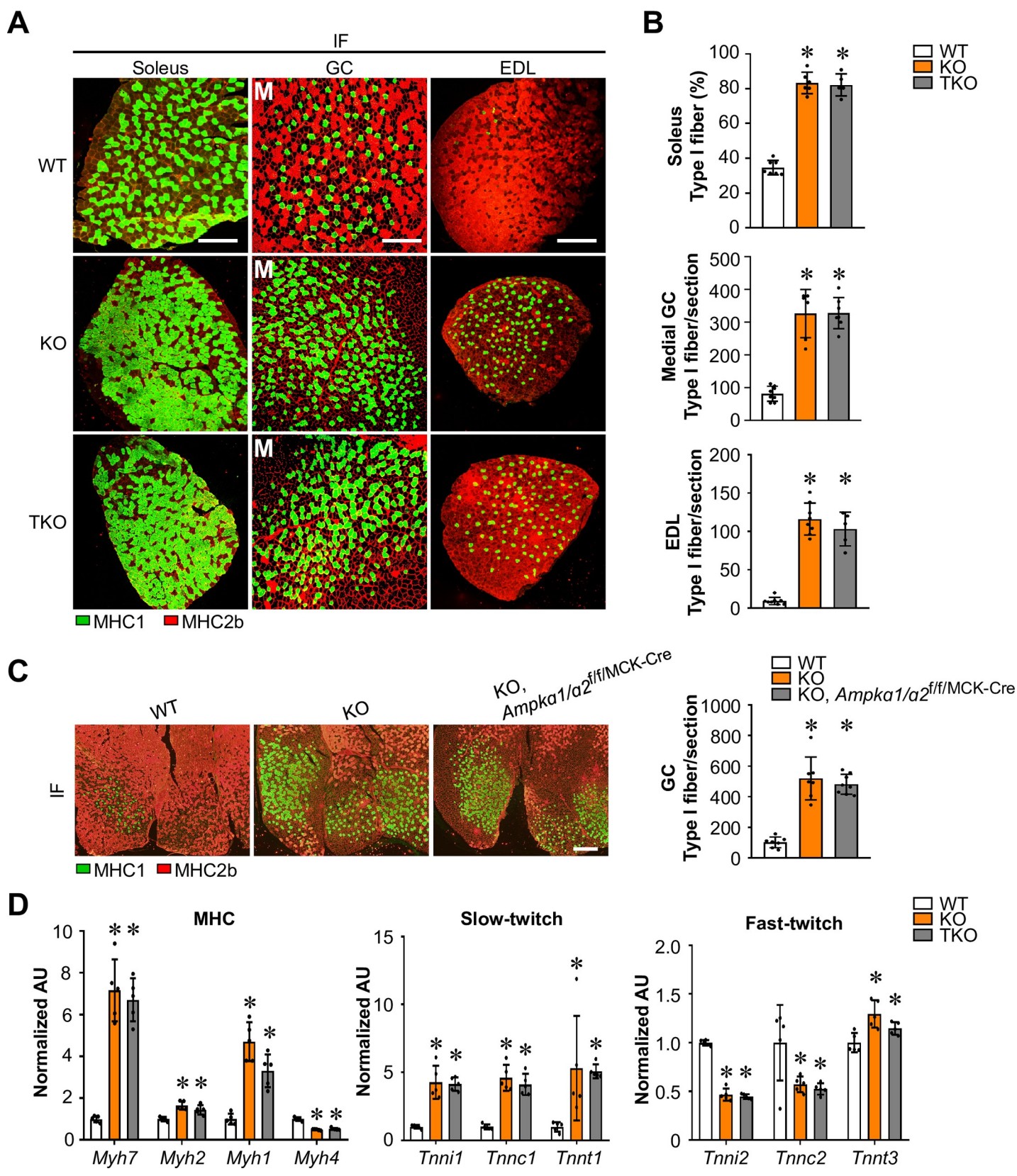

**Fig 6. AMPK-independent regulation of muscle fiber type switching by FNIP1.** (**A**) Cross-section of soleus, GC and EDL muscle from the indicated 8-9-week-old mice stained for MHC immunofluorescence (IF). Representative images were shown. M (medial head of GC). MHC1 (green), and MHC2b (red). Scale bar: 300 µm. (**B**) Quantification of IF data shown in (**A**). n = 5–8 mice per group. (**C**) (Left) Cross-section of GC muscle from the indicated 8-week-old WT, $Fnip1^{KO}$ and $Fnip1^{KO}$, $AMPKa1/a2^{f/f/MCK-Cre}$ mice stained for MHC immunofluorescence. Representative images were shown. MHC1 (green), and MHC2b (red). Scale bar: 300 µm. (Right) Quantification of IF data shown in (**C**). n = 7–8 mice per group. (**D**) Expression of MHC and representative slow/fast-twitch troponin genes (RT-qPCR) in entire GC muscle from indicated mice. n = 5 mice per group. Values represent mean ± SD, $^*P < 0.05$ vs. WT controls.

AMPK/PGC-1α, mitochondrial function and muscle fiber type are indeed controlled by distinct signaling pathways. Therefore, we have genetically established a pivotal intrinsic role for FNIP1 in orchestrating mitochondrial function and type I muscle fiber, as well as its epistatic relationship with AMPK and PGC-1α in driving type I muscle fiber determination in skeletal muscle.

Our data strongly suggest that the major effect of FNIP1 on muscle mitochondrial system might be mediated through inhibition of AMPK signaling. First, we found that AMPK but not mTORC1 was activated in $Fnip1^{KO}$ but reduced to basal levels in $Fnip1^{TgKO}$ muscle. Second, two independent triple knockout $Ampka1/\alpha2$ and $Fnip1$ mice experiments demonstrated that AMPK is required for the $Fnip1$ deficiency-mediated increase in mitochondrial oxidative program in skeletal muscle. Third, we confirmed that FNIP1 negatively regulates muscle cell mitochondrial program through inhibiting AMPK in a cell-autonomous setting. Notably, FLCN has also been shown to act through AMPK/PGC-1α function to handle various types of stress conditions in *C. elegans* and mammalian models [33–36], which further emphasizes the importance of AMPK/PGC-1α in mediating FNIP1/FLCN actions. However, a surprising finding of our study was dissociation between mitochondrial function and type I fiber program in TKO and DKO mice muscle, and we concluded that FNIP1-dependent type I muscle fiber program does not require the AMPK/PGC-1α signaling based on our multiple genetic mouse line studies. We show that disruption of $Ampka1/\alpha2$ in $Fnip1^{KO}$ muscle resulted in significantly reduced mitochondrial function but, surprisingly, have no effect on $Fnip1$ deficiency-mediated type I muscle fiber switching in multiple muscle types. Consistent with TKO and $Fnip1^{KO}$, $Ampka1/\alpha2^{f/f/MCK-Cre}$ mice, we found that FNIP1 is involved in type I muscle fiber determination independently from PGC-1α. It is possible independent pathways are activated in parallel with AMPK/PGC-1α signaling to regulate type I muscle fiber determination in $Fnip1^{KO}$ muscle. Previous studies have identified multiple transcriptional signaling, such as calcineurin/NFAT, HDAC/MEF2, nuclear receptor/miRNA axis, and HSF1 in the regulation of type I muscle fiber specification [25,37–41]. Interestingly, we found that the ERRs/miRNA axis is activated, whereas the expression of the genes encoding calcineurin, NFATs, MEF2s and HSF1 was not increased in $Fnip1^{KO}$ muscle. We have previously reported that $Fnip1$ is a direct target of miR-499, acting downstream of the ERRs/miR-499 axis [25,40], thereby suggesting a feedback loop whereby the FNIP1 pathways described here acts independently of PGC-1α to regulate fiber type specification.

We found that $Fnip1$ deficiency in skeletal muscle triggers a robust mitochondrial beneficial remodeling and glycolytic-to-oxidative muscle fiber transformation in the absence of exercise. Thus our findings reveal an exercise-independent FNIP1 pathway that coordinately regulates the mitochondrial and structural determinants of muscle fitness. Whereas we found that FNIP1 simultaneously regulates mitochondrial oxidative program and type I muscle fiber type in multiple muscle types across a range of fiber type proportions and oxidative capacity, our data also suggest that FNIP1 exerts a different control effect on MHC2a and MHC2x expression in different muscle types. This could reflect the intrinsic differences between muscles at the baseline that affect the adaptive range of MHC transformations [29]. Notably, the mitochondria in the skeletal muscle of $Fnip1^{KO}$ mice were more numerous and often significantly

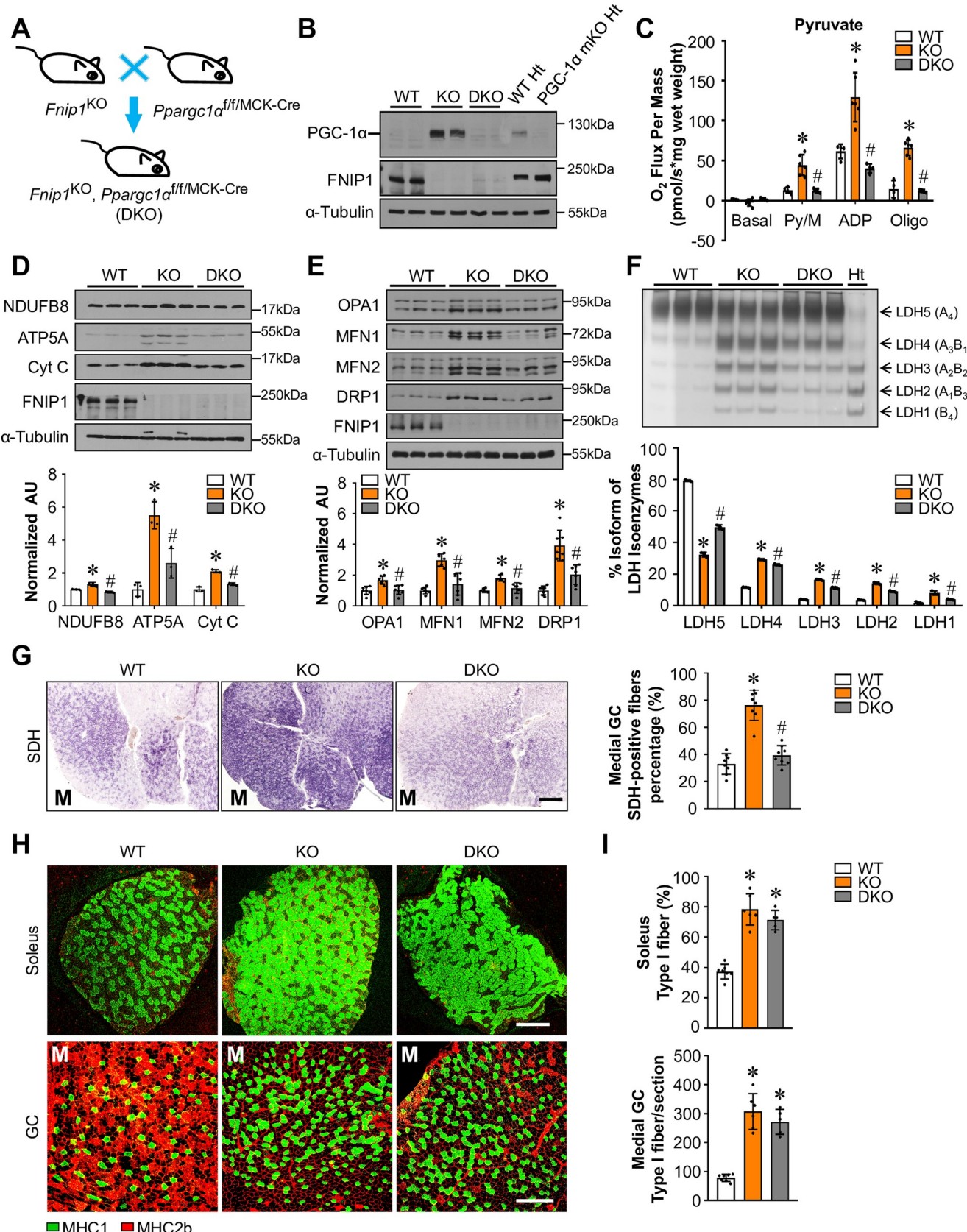

**Fig 7. PGC-1α is indispensable for the FNIP1-mediated control of muscle mitochondrial program but not fiber-type specification.** (**A**) Schematic showing the creation of *Fnip1*[KO], PGC-1α[f/f/MCK-Cre] (DKO) mice. (**B**) Representative immunoblot analysis performed with whole cell extracts from entire GC muscle and heart (Ht) from indicated mice using PGC-1α, FNIP1 and α-Tubulin (control) antibodies. n = 4 mice per group. (**C**) Mitochondrial respiration rates were determined from EDL muscle of the indicated genotypes using pyruvate as a substrate. Pyruvate/malate (Py/M)-stimulated, ADP-dependent respiration and oligomycin-induced (Oligo) are shown. n = 4–6 mice per group. (**D, E**) (Top) Representative immunoblot analysis of entire GC muscle lysates for mice of the indicated genotype using the indicated antibodies. (Bottom) Quantification of the NDUFB8/Tubulin, ATP5A/Tubulin, Cytochrome c/Tubulin, OPA1/Tubulin, MFN1/Tubulin, MFN2/Tubulin and DRP1/Tubulin signal ratios were normalized (= 1.0) to the WT controls. n = 3–7 mice per group. (**F**) (Top) LDH isoenzymes were separated by polyacrylamide gel electrophoresis using whole cell extracts from WV muscle from the indicated mice. A representative gel showing 3 independent mice per group is shown. (Bottom) Quantification of LDH isoenzyme activity gel electrophoresis was shown. (**G**) (Left) Cross-section of GC muscle from the indicated mice stained for SDH activity. Representative images were shown. M (medial head of GC). Scale bar: 300 μm. (Right) Quantification of SDH activity on the left. n = 8 mice per group. (**H**) Cross-section of soleus and GC muscle from the indicated mice stained for MHC immunofluorescence. Representative images were shown. M (medial head of GC). MHC1 (green), and MHC2b (red). Scale bar: 300 μm. (**I**) Quantification of IF data shown in (**H**). n = 5–7 mice per group. Values represent mean ± SD, *$P < 0.05$ vs. WT controls, # $P < 0.05$ vs. *Fnip1*[KO].

larger than the WT controls. In some muscle regions, the marked mitochondrial proliferation appeared to have replaced the myofibrillar contractile apparatus. In addition, our data also suggested a role of muscle FNIP1 in controlling body weight and muscle mass. It is possible that the difference in muscle size could reflect the muscle fiber type switching. It is also possible that FNIP1 directly regulates the dynamic balance between protein synthesis and degradation that determine muscle mass. Future studies will be necessary to further delineate the FNIP1 signaling in regulating muscle atrophy.

Whereas our results provide significant evidence that *Fnip1* deficiency triggers muscle mitochondrial function through activating AMPK, the mechanism whereby AMPK is activated in *Fnip1*[KO] muscle was not fully delineated in this study. Evidence has emerged that the lysosomal v-ATPase-ragulator complex is critical for AMPK activation [42]. Given that FNIP1 is known to be recruited to lysosomes [24,43], it is possible that FNIP1 could serve as an adaptor protein to interact with the AMPK activation complex at the surface of lysosomes, thus modulating the activation of AMPK. In addition, FNIP1 was recently shown to act as co-chaperone of Hsp90 to control protein stability [44]. Whether such mechanisms are relevant to the actions of FNIP1 on AMPK remains to be determined. In addition, our data suggest that FNIP1 could not only regulate mitochondria biogenesis, but also regulate mitochondrial quality. Consistent with this notion, recent study also showed that FNIP1 regulate mitochondrial redox homeostasis [45].

In summary, our study illustrates a previously unrecognized intrinsic role of FNIP1 that coordinately regulates mitochondrial function and muscle fiber type that govern muscle fitness. Given that many disease states are associated with diminished mitochondrial function and reduced numbers of slow muscle fibers. The regulatory network described here shows promise for the identification of therapeutic targets aimed at coordinately increasing mitochondrial function and muscle type I fibers.

## Materials and methods

### Ethics statement

All animal studies were conducted in strict accordance with the institutional guidelines for the humane treatment of animals and were approved by the IACUC committees at the Model Animal Research Center (MARC) of Nanjing University.

### Animal studies

WT C57BL/6J mice were from GemPharmatech Co., Ltd (Jiangsu, China). To generate mice with muscle-specific *Fnip1* overexpression (*Fnip1*[Tg]), a cDNA encoding the mouse *Fnip1* gene

was cloned into the EcoRV site downstream of the mouse *Mck* gene promoter (kind gift of E. N. Olson, University of Texas Southwestern). The transgene was linearized with XhoI and SacII digestion and microinjected into C57BL/6J embryos by the transgenic mouse facility at the Model Animal Research Center of Nanjing University. Transgenic mice were identified by PCR amplification of a 534-bp product using primers specific for *Fnip1* (5'-TTTCCAACC TGCTTCATTCCACTCTTCA) and the human growth hormone poly (A) component of the MCK construct (5'-ATTGCAGTGAGCCAAGATTGTGCCACTGCA). *Fnip1*[KO] mice were generated by the transgenic mouse facility at the Model Animal Research Center of Nanjing University. Briefly, the Cas9 system targeting the exon 6 of the C57BL/6J mice *Fnip1* gene was applied. The Cas9 protein and the sgRNAs were microinjected into the zygotes, and the injected zygotes were transplanted into the oviduct of pseudopregnant mothers. The sequences of sgRNA oligos are listed in S3 Table. *Fnip1*[Tg] mice were crossed with *Fnip1*[KO] mice to generate *Fnip1*[TgKO] mice expressing *Fnip1* only in skeletal muscle but not in other tissues. Generation of *Ampkα1/α2*[f/f] mice has been described elsewhere [46]. To generate mice with a muscle-specific disruption of the *Ampkα1/α2* alleles in *Fnip1*[KO] mice, *Ampkα1/α2*[f/f] mice were crossed with mice expressing Cre under control of the myogenic factor 5 (*Myf5*) promotor (Jackson Laboratory, stock no. 010529) or the muscle creatine kinase (*Mck*) promoter (Jackson Laboratory, stock no. 006475) to achieve muscle specific deletion of *Ampkα1/α2*, and these mice were then crossed with *Fnip1*[KO] mice to obtain *Fnip1*[KO], *Ampkα1/α2*[f/f/Myf5-Cre] (herein named TKO) or *Fnip1*[KO], *Ampkα1/α2*[f/f/MCK-Cre] mice. To generate mice with a muscle-specific disruption of the PGC-1α allele in *Fnip1*[KO] mice, PGC-1α[f/f] mice (Jackson Laboratory, stock no. 009666) were crossed with *Mck-Cre* to achieve muscle-specific deletion of PGC-1α and these mice were then crossed with *Fnip1*[KO] mice to obtain *Fnip1*[KO], PGC-1α[f/f/MCK-Cre] mice (herein named DKO). Mice at the age of 8 to 12 weeks were used. Mice were randomly assigned to various analyses. Investigators involved in the immunofluorescence imaging, RNA-seq, and histological analysis were blinded. Investigators performing animal handling, sampling, and raw data collection were not blinded.

## Mitochondrial respiration studies

Mitochondrial respiration rates were measured in saponin-permeablized extensor digital longus and plantaris muscle fibers with pyruvate or succinate as substrates as described previously [25,28,47]. In brief, the muscle fibers were separated and transferred to BIOPS buffer (7.23 mM $K_2EGTA$, 2.77 mM $CaK_2EGTA$, 20 mM imidazole, 20 mM taurine, 50 mM potassium 2-[N-morpholino]-ethanesulfonic acid, 0.5 mM dithiothreitol, 6.56 mM $MgCl_2$, 5.7 mM ATP, and 14.3 mM phosphocreatine [PCr], pH 7.1). The muscle fibers bundles were then permeabilized with 50 μg/ml saponin in BIOPS solution. Measurement of oxygen consumption in permeabilized muscle fibers was performed in buffer Z (105 mM potassium 2-[N-morpholino]-ethanesulfonic acid, 30 mM KCl, 10 mM $KH_2PO4$, 5 mM $MgCl_2$, 5 mg/ml BSA, 1 mM EGTA, pH7.4) at 37°C and in the oxygen concentration range 220–150 nmol $O_2$/ml in the respiration chambers of an Oxygraph 2K (Oroboros Inc., Innsbruck, Austria). Following measurement of basal, pyruvate (10 mM)/malate (5 mM) or succinate (5 mM)/rotenone (10 μM) respiration, maximal (ADP-stimulated) respiration was determined by exposing the mitochondria to 4 mM ADP. Uncoupled respiration was evaluated following addition of oligomycin (1 μg/mL). Respiration rates were determined and normalized to tissue wet weight using Datlab 5 software (Oroboros Inc., Innsbruck, Austria), the data were expressed as "pmol $O_2$ $s^{-1}$ mg wet weight$^{-1}$".

## Oxygen consumption measurements

Cellular oxygen consumption rates (OCR) and extracellular acidification rates (ECAR) were measured using the XF24 analyzer (Seahorse Bioscience Inc.) per the manufacturer's protocol

as described previously [25,28,47]. The basal OCR was first measured in XF Assay Media without sodium pyruvate, followed by administration of 10 mM sodium pyruvate. Uncoupled respiration was evaluated following the addition of oligomycin (2 μM) to inhibit ATP synthase, by addition of the uncoupler FCCP (2 μM), and then followed by the addition of rotenone/antimycin (1 μM). Immediately after measurement, total protein levels were measured with the Micro BCA Protein Assay Kit (Thermo scientific) for data correction.

## Histologic analyses

Muscle tissue was frozen in isopentane that had been cooled in liquid nitrogen. Notably, the gastrocnemius (GC) and plantaris of each mouse were dissected as a whole. 10 μm-thick serial GC muscles cross-sections were cut from the knee cut side in a Leica CM1850 cryostat at -20°C and mounted on positively charged glass slides. Transverse sections collected from the widest part (mid-belly) of the GC muscle were used for histological comparison to keep consistent between different mice. SDH and immunofluorescence (IF) stains were conducted as previously described [25,28,47]. For IF stains, the muscle fibers were stained with antibodies directed against MHC1 (BA-D5, #AB 2235587) or MHC2b (BF-F3, #AB 2266724). Briefly, slides were fixed in ice-cold 4% PFA for 5 min, and permeabilized with ice-cold 0.5% Triton X-100-PBS for 10 min. These sections were then blocked using 5% normal goat serum (NGS)-PBS for 30 min at room temperature, followed by incubation with anti-MHC2b antibody at 4°C overnight. Three consecutive washes with PBS for 5 min each were followed by sequential incubation with goat anti-mouse IgM Alexa 568 secondary antibody (Invitrogen A21043) for 1 hour at room temperature. Muscle sections were then washed three times with PBS for 5 min each, fixed in ice-cold 4% PFA for 2 min, washed with PBS for 5 min, followed by incubation with anti-MHC1 antibody for 3 hours at room temperature. Washed three times with PBS for 5 min each, followed by incubation with goat anti-mouse IgG Alexa 488 secondary antibody (Invitrogen A11001) for 1 hour at room temperature. The staining images were captured under the confocal microscope. Quantification of histological staining was performed in a blinded manner with Image-Pro Plus software. For type I fiber and SDH staining quantification in GC muscle, data were presented as number of type I fibers per medial head of GC or percentage of positive SDH staining myofibers per medial head of GC to keep consistent between different mice.

## RNA-Seq studies

Transcriptomics analyses were performed using RNA-sequencing as described previously [48,49]. Total RNA was isolated from the entire gastrocnemius muscle of 8-week-old male *Fnip1*<sup>KO</sup>, *Fnip1*<sup>TgKO</sup> and WT control mice using RNAiso Plus (Takara Bio). RNA-seq using Illumina HiSeq 4000 was performed by Beijing Novogene Bioinformatics Technology Co., Ltd. Two independent samples per group were analyzed. Paired-end, 150 nt reads were obtained from the same sequencing lane. Transcriptome sequencing libraries averaged 39 million paired reads per sample, with 87.1% alignment to the mouse genome (UCSC mm10). The sequencing reads were then aligned to the UCSC mm10 genome assembly using TopHat 2.0.14 with the default parameters. Fragments Per Kb of exon per Million mapped reads (FPKM) were calculated using Cufflinks 2.2.1. The criteria for a regulated gene were a fold change greater than 1.5 (either direction) and a significant P-value ($< 0.05$) versus WT. For pathway analysis, the filtered data sets were uploaded into DAVID Bioinformatics Resources 6.8 to review the bio pathways using the Functional Categories database. The GO analysis was used to interpret data, and the regulated terms were ranked by P-value. The heat-map analysis of regulated genes was generated by using R software (Version 3.3.2) and ggplot2/gplots

package. The RNA-seq data have been deposited in the NCBI Gene Expression Omnibus and are accessible through GEO Series accession number GSE150193.

## RNA analyses

Quantitative RT-PCR was performed as described previously [47,48]. Briefly, total RNA was extracted from entire gastrocnemius or soleus muscle using RNAiso Plus (Takara Bio). The purified RNA samples were then reverse transcribed using the PrimeScript RT Reagent Kit with gDNA Eraser (Takara Bio). Real-time quantitative RT-PCR was performed using the ABI Prism Step-One system with Reagent Kit from Takara Bio. Specific oligonucleotide primers for target gene sequences are listed in S4 Table. Arbitrary units of target mRNA were corrected to the expression of *36b4*.

## Transmission electron microscopy

Mice were euthanized and perfused with sodium phosphate buffer (PB, 100mM, pH7.4) and pre-fixed solution (2.5% (vol/vol) glutaraldehyde, 1% paraformaldehyde in PB). Tibialis anterior (TA) muscle was dissected, cut into small pieces and fixed in the same pre-fixed solution overnight at 4°C. After rinsing with PB, tissues were immersed in 0.2 M imidazole in PB for 15 minutes, and then post-fixed with 1% osmium tetraoxide in PB. After rinsing with high-purity water, the samples were stained with 1% aqueous lead at 4°C overnight. Gradient dehydration was accomplished by incrementing the concentration of acetone and embedded in epoxy resin (60°C for 24 h). Samples were sectioned using Leica EM UC7 and placed on copper grids. Images were taken on a FEI Tecnai G2 20 Twin electron microscope equipped with an Eagle 4k CCD digital camera (FEI; USA) in a double-blind manner.

## Mitochondrial DNA analyses

Genomic/mitochondrial DNA was measured as described previously [25]. Mitochondrial DNA content was determined by SYBR Green analysis (Takara Bio). The levels of NADH dehydrogenase subunit 1 (mitochondrial DNA) were normalized to the levels of lipoprotein lipase (genomic DNA). The primer sequences are noted in S4 Table.

## Visualization of mitochondria

The number of mitochondria and their distribution were determined using the MitoTracker Red CMXRos selective probe (Invitrogen catalog #M7512). The myotubes were incubated in a medium without FBS and containing 50 nM MitoTracker for 30 min at 37°C. The medium was then replaced with a complete medium but without MitoTracker. Live cell images were acquired using the LEICA DFC450 Imaging System. All samples were analysed using the same settings. The quantifications of mitotracker fluorescent signal intensity were performed in a blinded manner with Image J Fiji software.

## Antibodies and immunoblotting studies

Antibodies directed against MHC1 (BA-D5) and MHC2b (BF-F3) were purchased from the Developmental Studies Hybridoma Bank; antibodies directed against NDUFB8 (14794-1-AP, 1:1,000 dilution), UQCRC2 (14742-1-AP, 1:1,000 dilution), COX4 (11242-1-AP, 1:1,000 dilution), ATP5A (14676-1-AP, 1:1,000 dilution), MFN1 (13798-1-AP, 1:1,000 dilution), DRP1 (12957-1-AP, 1,1,000 dilution) were all from Proteintech; antibodies directed against OPA1 (612606, 1,1,000 dilution) were from BD Biosciences; antibodies directed against cytochrome c (bs1089, 1:1,000 dilution) and α-tubulin (bs1699, 1:5,000 dilution) were from Bioworld;

antibodies directed against myoglobin (sc-25607, 1:1,000 dilution), MFN2 (sc-100560, 1:500 dilution) were from Santa Cruz; antibodies directed against FLCN (#3697, 1:1,000 dilution), p-AMPKα Thr172 (#2535, 1:1,000 dilution), AMPKα (#5831, 1:1,000 dilution), p-ACC Ser79 (#11818, 1:1,000 dilution), ACC (#3676, 1:1,000 dilution), p-mTOR Ser2448 (#5536, 1:1,000 dilution), p-mTOR Ser2481 (#2974, 1:1,000 dilution), mTOR (#2983, 1:1,000 dilution), p-S6K (#9234, 1:1,000 dilution), S6K (#2708, 1:1,000 dilution), p-S6 (#4858, 1:1,000 dilution), S6 (#2317, 1:1,000 dilution), p-4EBP1 (#2855, 1:1,000 dilution) and 4EBP1 (#9644, 1:1,000 dilution) were from Cell Signaling Technology; antibody directed against β-Actin (AC026, 1:5,000 dilution) was from Abclonal; anti-PGC-1α (1,1,000 dilution) was developed in the laboratory of Daniel Kelly as previously described [50]; anti-FNIP1 was developed in the laboratory of Zhenji Gan with the help with Abcam (ab236547, 1:500 dilution); Western blotting studies were performed as previously described [47,48]. Blots were normalized to α-tubulin. The total protein concentration was measure by BCA assay using Pierce BCA Assay Kit Protocol (ThermoFischer Scientific). Equal total protein was loaded to each lane and the membranes were also stained with 0.1% Ponceau S.

## LDH isoenzyme analysis and activity assay

LDH isoenzyme patterns were determined as previously described [28]. Briefly, mouse skeletal muscle were homogenized in a solution of 0.9% NaCl, 5 mM Tris-HCl, pH 7.4, and the lysates were centrifuged for 30 min at $15,000 \times g$ to remove the cellular debris. 100 μg of protein/lane was loaded onto a 6% nondenaturing polyacrylamide gel. Following electrophoresis, the gel was placed in 10 ml of staining solution containing 0.1 M sodium lactate, 1.5 mM NAD, 0.1 M Tris-HCl (pH 8.6), 10 mM NaCl, 5 mM MgCl2, 0.03 mg/ml phenazinmethosulphate, and 0.25 mg/ml nitro blue tetrazolium. Protein extracted from mouse heart served as a positive control.

## Cell culture

Primary muscle cells were isolated from gastrocnemius (GC) muscles of 4-week old male mice as previously described [25,40,47,51]. Briefly, GC muscles from both legs were removed. Minced tissue was digested in a collagenase/dispase/CaCl$_2$ solution for 1.5 hours at 37°C in a shaking bath. DMEM supplemented with 10% FBS (PPM) was added and samples were triturated gently before loading onto a Netwell filter (70 μm, BD). Cell suspension was pelleted at 1000 rpm for 5 minutes. Cells were then resuspended in PPM and plated on an uncoated plate for differential plating. Cell suspension (not-adherent) was centrifuged for 5 minutes at 1000 rpm and pellet was resuspended in Growth Medium (GM) (Ham's F-10 medium supplemented with 20% FBS and 2.5 ng/ml bFGF). Cells were plated on collagen coated flasks for expansion. Cells were fed daily with GM. For differentiation, cells were washed with PBS and re-fed with 2% Horse-serum/DMEM differentiation medium and re-fed daily. Cells were induced to differentiation for 3 days prior to various experiments.

## Statistical analyses

All mouse and cell studies were analyzed by Student's *t* test when two groups were compared. One-way ANOVA coupled to a Fisher's least significant difference (LSD) post-hoc test or two-way ANOVA coupled to Bonferroni post-hoc test were used when more than two groups were compared. Data represent the mean ± SD, with a statistically significant difference defined as a value of $P < 0.05$. All the numerical data were presented in S1 Data.

## Supporting information

**S1 Fig. Generation and characterization of muscle-specific *Fnip1* transgenic mice.** (**A**) The schematic depicts the *Mck-Fnip1* construct used for *Fnip1*<sup>Tg</sup> transgene production. (**B**) RT-qPCR analysis of *Fnip1* mRNA levels in the entire gastrocnemius (GC) muscle from nontrans-genic (NTG) and *Fnip1*<sup>Tg</sup> mice. n = 4 mice per group. (**C**) Representative immunoblot analysis of protein extracts prepared from white vastus lateralis (WV) muscles, heart (Ht) and liver of the indicated mice using FLAG and α-Tubulin (control) antibodies. n = 4 mice per group. (**D**) (Left) Pictures of NTG and *Fnip1*<sup>Tg</sup> mice at the age of 8 weeks. (Right) Representative soleus, GC and WV muscles from indicated mice. (**E**) Cross-section of GC muscle from 8-week-old male NTG and *Fnip1*<sup>Tg</sup> mice stained for MHC immunofluorescence (IF). Representative images were shown. M (medial head of GC), MHC1 (green), and MHC2b (red). Scale bar: 300 μm. (**F**) Quantification of IF data shown in (**E**). n = 3–4 mice per group. Values represent mean ± SD, *$P < 0.05$ vs. NTG controls.
(TIF)

**S2 Fig. Generation and characterization of *Fnip1* KO mice.** (**A**) Schematic showing the Cas9 system targeting exon 6 of the *Fnip1* gene to generate *Fnip1* knockout mice (*Fnip1*<sup>KO</sup>). (**B**) Detection of the *Fnip1* mutation by PCR. Primers flanking the exon 6 of the *Fnip1* gene gener-ate PCR products as indicated. (**C**) Results of RT-qPCR analysis of *Fnip1*, *Fnip2* and *Flcn* mRNA levels in entire GC muscles from indicated mice. n = 5 mice per group. (**D**) Immuno-blot analysis of protein extracts prepared from WV muscles of the indicated mice using FNIP1, FLCN and α-Tubulin (control) antibodies. n = 4 mice per group. (**E**) Representative pictures showing WT and *Fnip1*<sup>KO</sup> mice skeletal muscle at the age of 8 weeks. Values represent mean ± SD, *$P < 0.05$ vs. WT controls.
(TIF)

**S3 Fig. Analysis of genes regulated in *Fnip1*<sup>KO</sup> muscle.** (**A**) Volcano plot showing fold changes versus *P*-values for the analyzed RNA-seq data generated from the entire GC muscle of 8-week-old male *Fnip1*<sup>KO</sup> mice compared to WT controls. Significantly up-regulated genes are represented by red dots, whereas down-regulated genes are represented by blue dots. (**B**) Gene ontology (GO) enrichment analysis of gene transcripts (top 3000) up-regulated in *Fnip1*<sup>KO</sup> muscle, with the top ten terms shown. (**C**) Heat-map of up-regulated mitochondrial complex genes in *Fnip1*<sup>KO</sup> muscles, color scheme for fold change is provided. (**D**) GO enrich-ment analysis of 739 gene transcripts down-regulated in *Fnip1*<sup>KO</sup> muscle. (**E**) Heat maps depicting a subset of gene expression data, individual genes involved in the regulation of type I muscle fiber specification are shown to be regulated in the *Fnip1*<sup>KO</sup> as denoted by the color scheme (relative fold change compared to WT control shown to the right of each gene). (**F**) RNA-seq data of WT and *Fnip1*<sup>KO</sup> muscle indicate an increase expression of miR-208b and miR-499.
(TIF)

**S4 Fig. Analysis of genes regulated in *Fnip1*<sup>TgKO</sup> muscle.** (**A**) Volcano plot showing fold changes versus *P*-values for the analyzed RNA-seq data generated from the entire GC muscle of 8-week-old male *Fnip1*<sup>TgKO</sup> mice compared to WT controls. Significantly up-regulated genes are represented by red dots, whereas down-regulated genes are represented by blue dots. (**B**, **C**) GO enrichment analysis of gene transcripts up-regulated in *Fnip1*<sup>TgKO</sup> muscle. (**D**, **E**) GO enrichment analysis of gene transcripts down-regulated in *Fnip1*<sup>TgKO</sup> muscle.
(TIF)

**S5 Fig. FNIP1-dependent muscle oxidative transformation.** (**A**, **C**) Immunoblot analysis of entire GC muscle lysates for mice of the indicated genotype using the indicated antibodies. Ponceau red staining was also shown. n = 3 mice per group. (**B**) Mitochondrial respiration rates were determined from the plantaris part of the gastrocnemius/plantaris complex of the indicated genotypes using pyruvate or succinate as substrates. Pyruvate/malate (Py/M) or succinate/rotenone (Suc/Rot)-stimulated, ADP-dependent respiration and oligomycin-induced (Oligo) are shown. n = 4 mice per group. (**D**) (Top) Immunoblot analysis of WV muscle lysates for mice of the indicated genotype using the myoglobin antibody. (Bottom) Ponceau red staining was shown. n = 3 mice per group. (**E**) Quantification of LDH isoenzyme activity gel electrophoresis. Values represent the mean % (+/- SD) total LDH activity. n = 3 mice per group. Values represent mean ± SD, $^*P < 0.05$ vs. WT controls, # $P < 0.05$ vs. $Fnip1^{KO}$. (TIF)

**S6 Fig. mTOR signaling-independent regulation by FNIP1.** (**A**) Low power MHC immunofluorescence scan showing entire cross-section of the mid-belly GC muscle of WT, $Fnip1^{KO}$ and $Fnip1^{TgKO}$ mice at the age of 8 weeks. (M) medial; (L) lateral; (PL) plantaris. MHC1 (green), and MHC2b (red). Scale bar: 1000 μm. (**B**) Immunoblot analysis of WV muscle lysates for mice of the indicated genotype using the indicated antibodies. (**C**) Quantification of the p-S6K (Thr389)/S6K, p-S6 (Ser235/236)/S6 and p-4EBP1 (Thr37/46)/4EBP1 signal ratios were normalized (= 1.0) to the $Fnip1^{KO}$. n = 3 mice per group. (TIF)

**S7 Fig. FNIP1 regulates muscle mitochondrial oxidative programs through AMPK.** (**A**) (Top) Immunoblot analysis of muscle lysates for mice of the indicated genotype using the cytochrome c antibody. (Bottom) Ponceau red staining was shown. n = 3 mice per group. (**B-D**) Primary skeletal muscle myoblasts were isolated from GC muscles of WT, $Fnip1^{KO}$, or $AMPKa1/a2^{f/f/Myf5-Cre}$ (TKO) mice, and then induced to differentiation for 3 days. (**B**) (Top) Immunoblot analysis of myotube extracts using the COX4 antibody. (Bottom) Ponceau red staining was shown. (**C**) (Top) Myotubes were stained with MitoTracker Red CMXRos, (Bottom) Phase contrast microscopy images of myotubes were shown. Scale bars: 100 μm. (**D**) Quantification of mitotracker fluorescent signal intensity in (**C**). n = 3 independent experiments. Values represent mean ± SD, $^*P < 0.05$ vs. WT controls, # $P < 0.05$ vs. $Fnip1^{KO}$. (TIF)

**S8 Fig. AMPK-independent regulation of muscle fiber type switching by FNIP1.** Expression of myosin heavy chain (MHC) and representative slow/fast-twitch troponin genes (qRT-PCR) in soleus muscles from indicated mice. n = 4–6 mice per group. Values represent mean ± SD, $^*P < 0.05$ vs. WT controls. (TIF)

**S9 Fig. PGC-1α is indispensable for the FNIP1-mediated control of muscle mitochondrial oxidative program.** (Top) Immunoblot analysis of entire GC muscle lysates for mice of the indicated genotype using the cytochrome c antibody. (Bottom) Ponceau red staining was shown. n = 3 mice per group. (TIF)

**S1 Table. Body weight and muscle weight measurements.** (DOCX)

**S2 Table. FNIP1-dependent mitochondrial-related genes (455).** (DOCX)

**S3 Table. sgRNA for *Fnip1* knockout mice generation.**
(DOCX)

**S4 Table. RT-PCR primers.**
(DOCX)

**S1 Data. Raw numerical data of all the figures.**
(XLSX)

## Author Contributions

**Conceptualization:** Liwei Xiao, Jing Liu, Zongchao Sun, Zhenji Gan.

**Data curation:** Liwei Xiao, Jing Liu, Zongchao Sun, Zhenji Gan.

**Formal analysis:** Liwei Xiao, Jing Liu, Zongchao Sun.

**Funding acquisition:** Tingting Fu, Zhenji Gan.

**Investigation:** Liwei Xiao, Jing Liu, Zongchao Sun, Yujing Yin, Yan Mao, Dengqiu Xu, Lin Liu, Zhisheng Xu, Qiqi Guo, Chenyun Ding, Wanping Sun, Likun Yang, Zheng Zhou, Danxia Zhou, Tingting Fu, Wenjing Zhou, Yuangang Zhu.

**Resources:** Xiao-Wei Chen, John Zhong Li, Shuai Chen, Xiaoduo Xie.

**Supervision:** Zhenji Gan.

**Validation:** Liwei Xiao, Jing Liu, Zongchao Sun.

**Visualization:** Liwei Xiao, Jing Liu, Zongchao Sun.

**Writing – original draft:** Liwei Xiao, Zhenji Gan.

**Writing – review & editing:** Liwei Xiao, Zongchao Sun, Zhenji Gan.

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
