## [Decision Letter · Decision Letter 0]

13 Oct 2020

Dear Dr Gan,

Thank you very much for submitting your Research Article entitled 'AMPK-dependent regulation of skeletal muscle mitochondrial oxidative program by FNIP1' to PLOS Genetics. Your manuscript was fully evaluated at the editorial level and by independent peer reviewers. The reviewers appreciated the attention to an important problem, but raised some substantial concerns about the current manuscript. Based on the reviews, we will not be able to accept this version of the manuscript, but we would be willing to review again a much-revised version. We cannot, of course, promise publication at that time. We are sorry that the review process took much mmore time than usually anticipated. We are specifically concerned by the Reviewer 2 comments about the novelty of this work, so please address this in details. In additiona, we expect you to addrress Reviewer 1 and 3 comments in full. 

If you decide to revise the manuscript for further consideration at PLOS Genetics, please aim to resubmit within the next 60 days, unless it will take extra time to address the concerns of the reviewers, in which case we would appreciate an expected resubmission date by email to plosgenetics@plos.org.

[LINK]

We are sorry that we cannot be more positive about your manuscript at this stage. Please do not hesitate to contact us if you have any concerns or questions.

Yours sincerely,

Aleksandra Trifunovic

Associate Editor

PLOS Genetics

Gregory Barsh

Editor-in-Chief

PLOS Genetics

Reviewer's Responses to Questions

**Comments to the Authors:**

Reviewer #1: In this massive work, Xiao and colleagues elegantly dissected the role of FNIP1 in the control of mitochondrial biogenesis and muscle fibres switching by using a series of genetically modified mouse and cellular models. Intriguingly, the authors found that while the effect on mitochondrial biogenesis is unsurprisingly mediated by the AMPK/PGC1alpha pathway, the effect on fibres type does not involve AMPK/PGC1alpha, although the mechanism of the latter is not further investigated. Importantly, the authors also demostarte that mTORC1 signalling is not involved in FNIP1-dependent activities.

The paper is convincing and the conclusions supported by the data.

However, there are some comments I would like the authors to consider.

1) the title is somehow misleading as it does not mention what is, in my opinion, the most relevant finding, i.e. the fact that AMPK is not involved in the determination of the fibre type by FNIP1.

2) in both abstract and authors' summary, the authors stress the fact that FNIP1 can be a novel target for the therapy of a number of diseases. I find this statement generic and completely out of the main subject of the paper, and suggest the authors to remove the sentences.

3) in figure S2C FLNC and FNIP2 are shown, but no explanation of what they are is given in the main text, although it is present in the Discussion. However, I suggest to move the explanation to the main text for clarity. In particular, it is unclear what FNIP2 is.

4) Figure 1F, sowing the results from the TEM studies should be properly quantified, and the quality of the images improved as it is difficult to clearly distinguish the mitochondrial ultrastructure.

5) In the expression profile 27/34 genes upregulated in the FNIP1 KO muscles are from respiratory complex I, but in the western blot in Figure 2A all the complexes are increased. Are the corresponding transcripts also increased?

6) why are myoglobin and cytochrome c upregulated in FNIP1 KO (Figure 2B)? No comment or discussion is made on this point.

7) in Figure 2C and 2D the authors show a switch in LDH isoforms, however it is unclear if this impacts on the blood lactate levels.

8) Page 11 line 233 Fnip1KO induced... should be changed to "increased"

Reviewer #2: This paper does not advance the field. The effect of FNIP1 KO in skeletal muscle has been published in 2015 by Reyes et al. The extra data showing that the fiber typ switch is not dependent on AMPK and PGC-1a is not conclusive. It is against Hasumi et al, 2012, who show convincingly that the red muscle fibre phenotype is reversed upon PGC1a KO.

Reviewer #3: The authors investigated the role of folliculin interacting protein 1 (Fnip1) in skeletal muscle phenotypes using transgenic and muscle-specific knockout mice. They report that Fnip1-KO induced a glycolytic-to-oxidative shift of the muscular metabolic profile via AMPK activation. Various experimental approaches were used, and the results are clear at a glance. The design of the experiments, however, has serious concerns.

The authors analyzed various indexes using several kinds of hindlimb skeletal muscles. The phenotypes of mammalian skeletal muscles vary: fast-twitch and glycolytic muscle fibers dominate in the lateral gastrocnemius, EDL, and vastus lateralis muscles; slow-twitch oxidative fibers dominate in the solus and medial gastrocnemius (GAS) muscles; and fast and glycolytic fibers dominate in the TA muscle, although slow and oxidative fibers dominate in the deep portion. The role of Fnip1 may therefore differ from one mammalian muscle type to another.

The volume of skeletal muscle available for the experiments may have fallen short of the authors’ requirements. The authors wanted to analyze various indexes using many kinds of skeletal muscle. The responses to Fnip1-KO in fast-type muscle, however, may differ from those in slow-type muscle, given that the nervous and endocrine systems may also mediate the muscle fiber type. The authors should analyze various indexes using the same kind of skeletal muscle.

Many of the analyses were carried out using the GAS muscle. This GAS is big, hence the contractile and metabolic properties of the muscle depend on the region examined. The sizes of the sampled muscles and the actual portions of the samples are left unstated in the manuscript. The paper should clearly describe the region of the GAS muscle samples in three dimensions.

Many studies have been done on the regulatory mechanisms of myosin heavy chain (MyHC) phenotypes. Previous reports have proposed that MyHC is mediated not only by PGC1-a and MAPK, but also several molecules such as calcineurin, the NFAT family, miR-208b, miR-499, and HSF1. The authors should discuss the relationship between Fnip1-AMPK signals and these candidates.

Figure 1F:

The images of Fnip1-KO are strange. Very few myofibrils appear in the upper image. This image may be derived from around the subsarcolemmal region, not from the intermyofibrils. And the mitochondria in both images appear to be swelling, with wide interstitial spaces. These are abnormal ultrastructures that may be attributed to technical errors during the fixation steps. The authors should replace the images with better-quality alternatives.

Figures 2E and 2F:

As mentioned above, the contractile and metabolic properties depend on the regions of GAS examined. Both the size of the sampled muscles and the actual portions of the samples are left unstated. The paper should clearly describe the region of the GAS muscle samples in three dimensions.

Judging from the immunostained images, the percent proportion of type I fibers in the soleus was underestimated in the graph. The proportions of type I soleus fibers in the wild-type, KO, and TgKO mice appear to be ~80%, ~100% and ~90%, respectively. And, how did the authors calculate the proportions of type I fibers and SDH-positive fibers in the GAS? As mentioned above, the contractile and metabolic properties depend on the regions of the GAS examined.

These figures also report muscle fiber atrophy in the KO mice, compared to the WT and TgKO. Why was this? The authors should validate their data by reporting both the body weights and the absolute and relative muscle weights.

Many unstained fibers appear in the images. Were these type 2a? If so, the authors should explain the effects of Fnip1-KO on type 2a fibers. Note also that the effects of Fnip1-KO may vary among muscle types.

Figures 3F and 5D:

Why did the authors evaluate the mitochondrial density using myoblasts instead of myotubes? All of the other analyses were performed using myotubes. Phase-contrast images should be shown.

Figures 3G and 2H:

The origin of the primary cells should be shown.

Figures 6A and 7H:

Judging from the images, Fnip1-KO induced muscle atrophy in the soleus and EDL in Figure 6A and in the GAS in Figure 7H. The results vary from figure to figure. The authors should explain this variation. Showing a cross-sectional area of the fiber in each condition would help to validate the data.

According to the figures, Fnip1-KO changed the proportion of unstained fibers. Why was this?

And again, as mentioned, the contractile and metabolic properties depend on the regions of the GAS examined. The manuscript offers no description of either the sizes of the sampled muscles or the actual portions of the samples. The paper should clearly describe the region of the GAS muscle samples in three dimensions.

Figure 7B:

The authors failed to detect PGC-1a by Western blotting. And, what does “n.s.” stand for here?

The authors should provide a detailed description of the procedure for their primary cell culture experiments. How long were the cells incubated with the differentiation medium? Note that the differentiation level may affect the metabolic profiles.

Figures 3G and 5E:

What kind of analysis was applied for the statistical analyses? This type of data should be analyzed using 2-way ANOVA.

Lines 130-131, 136-138:

At this point, none of the experimental data support a glycolytic-to-oxidative shift of skeletal muscles in Fnip1-KO mice. The result merely shows a shift in the color of the muscle from whitish to red.

Line 133:

“FLCN” should be spelled out.

“IF” should be spelled out.

a-tubulin is unsuitable for the loading and transfer control of Western blotting.

**Have all data underlying the figures and results presented in the manuscript been provided?**

Reviewer #1: Yes

Reviewer #2: Yes

Reviewer #3: Yes

PLOS authors have the option to publish the peer review history of their article (what does this mean?). If published, this will include your full peer review and any attached files.

Reviewer #1: No

Reviewer #2: No

Reviewer #3: No

---

## [Decision Letter · Decision Letter 1]

25 Jan 2021

Dear Dr Gan,

Thank you very much for submitting your Research Article entitled 'AMPK-dependent and -independent coordination of mitochondrial function and muscle fiber type by FNIP1' to PLOS Genetics.

The manuscript was fully evaluated at the editorial level and by independent peer reviewers. The reviewers appreciated the attention to an important problem, but still raised some substantial concerns about the current manuscript. Based on the reviews, we will not be able to accept this version of the manuscript, but we would be willing to review a revised version. We cannot, of course, promise publication at that time.

If you decide to revise the manuscript for further consideration at PLOS Genetics, please aim to resubmit within the next 60 days, unless it will take extra time to address the concerns of the reviewers, in which case we would appreciate an expected resubmission date by email to plosgenetics@plos.org.

[LINK]

We are sorry that we cannot be more positive about your manuscript at this stage. Please do not hesitate to contact us if you have any concerns or questions.

Yours sincerely,

Aleksandra Trifunovic

Associate Editor

PLOS Genetics

Gregory Barsh

Editor-in-Chief

PLOS Genetics

Reviewer's Responses to Questions

**Comments to the Authors:**

Reviewer #1: The authors addressed all my concerns, and made a relevant effort to improve the manuscript. I don't have any further question at this point.

Reviewer #3: The authors have revised their manuscript in response to my previous comments. Though several points of concern have been addressed in the revised manuscript, a few others still require attention.

Figure 1F:

As I pointed out, the EM images are unusual. The new images added as replacements, moreover, are of poor quality.

The replacement images of Fnip1-KO are still unusual. Very few myofibrils appear in the upper image, and those that do are atrophied. If the myofibrils are in fact as scarce and atrophied as they appear, the skeletal muscle may develop very low force, if any. The width of the I-band, moreover, is too narrow, and the mitochondria appear to be swelling, with wide interstitial spaces.

The new images of TgKO are also unusual, showing shortened sarcomeres. These are abnormal ultra-structures that may have resulted from technical errors during the fixation steps. The authors should replace the images with better-quality alternatives.

Figures 3F and 5D:

The authors added MitoTracker-stained images of myotubes as the new Fig. S7. They did not, however, make any evaluations on the mitochondrial density of these images. As I pointed out, the mitochondrial density should be assessed using myotubes instead of myoblasts. What, specifically, does the “higher mitochondrial potential” in in the myoblasts describe? Myoblasts cannot be representative of myofibers.

The authors added a Ponceau-stained image as a transfer and internal control for Western blotting in only one figure, Fig. S5C. Do the other Western blotting images need Ponceau-stained image?

**Have all data underlying the figures and results presented in the manuscript been provided?**

Reviewer #1: Yes

Reviewer #3: None

PLOS authors have the option to publish the peer review history of their article (what does this mean?). If published, this will include your full peer review and any attached files.

Reviewer #1: No

Reviewer #3: No

---

## [Editor Report · Decision Letter 2]

12 Mar 2021

Dear Dr Gan,

We are pleased to inform you that your manuscript entitled "AMPK-dependent and -independent coordination of mitochondrial function and muscle fiber type by FNIP1" has been editorially accepted for publication in PLOS Genetics. Congratulations!

Yours sincerely,

Aleksandra Trifunovic

Associate Editor

PLOS Genetics

Gregory Barsh

Editor-in-Chief

PLOS Genetics

Comments from the reviewers (if applicable):

**Data Deposition**

http://datadryad.org/submit?journalID=pgenetics&manu=PGENETICS-D-20-01073R2

**Press Queries**

---

## [Editor Report · Acceptance letter]

24 Mar 2021

PGENETICS-D-20-01073R2 

AMPK-dependent and -independent coordination of mitochondrial function and muscle fiber type by FNIP1 

Dear Dr Gan, 

We are pleased to inform you that your manuscript entitled "AMPK-dependent and -independent coordination of mitochondrial function and muscle fiber type by FNIP1" has been formally accepted for publication in PLOS Genetics! Your manuscript is now with our production department and you will be notified of the publication date in due course.

With kind regards,

Andrea Szabo

PLOS Genetics

On behalf of:
